# Development of a Virtual Reality Escape Room Game for Emotion Elicitation



Inês Oliveira [1,*], Vítor Carvalho [1,2], Filomena Soares [2], Paulo Novais [2], Eva Oliveira [1] and Lisa Gomes [3,*]

1   2Ai, School of Technology, Polytechnic Institute of Cavado and Ave, 4750-810 Barcelos, Portugal; vcarvalho@ipca.pt (V.C.); eoliveira@ipca.pt (E.O.)
2   Algoritmi Research Center/LASI, University of Minho, 4800-058 Guimaraes, Portugal; fsoares@dei.uminho.pt (F.S.); pjon@di.uminho.pt (P.N.)
3   School of Nursing, University of Minho, 4710-057 Braga, Portugal
*   Correspondence: ines.t.oliveira@gmail.com (I.O.); lgomes@ese.uminho.pt (L.G.)

**Abstract:** In recent years, the role of emotions in digital games has gained prominence. Studies confirm emotions' substantial impact on gaming, influencing interactions, effectiveness, efficiency, and satisfaction. Combining gaming dynamics, Virtual Reality (VR) and the immersive Escape Room genre offers a potent avenue through which to evoke emotions and create a captivating player experience. The primary objective of this study is to explore VR game design specifically for the elicitation of emotions, in combination with the Escape Room genre. We also seek to understand how players perceive and respond to emotional stimuli within the game. Our study involved two distinct groups of participants: Nursing and Games. We employed a questionnaire to collect data on emotions experienced by participants, the game elements triggering these emotions, and their overall user experience. This study demonstrates the potential of VR technology and the Escape Room genre as a powerful means of eliciting emotions in players. "Escape VR: The Guilt" serves as a successful example of how immersive VR gaming can evoke emotions and captivate players.

**Keywords:** emotions; triggers; Virtual Reality; games; Escape Room





## 1. Introduction

### 1.1. Context and Importance

In recent decades, studies on emotions have emerged in various fields, ranging from diverse scientific disciplines to artistic and philosophical areas. One of the scientific domains that has been exploring emotions is digital games [1,2]. Currently, games are a highly popular activity that provide leisure, entertainment, and learning to their users [3–7]. Due to recent research, game developers have started to pay attention to the player's emotional experience during a game, aiming to engage players as deeply as possible in the experience [1,3,5].

In order for the elicitation of emotions to be effective in a game, it is necessary to understand how emotions occur in human nature [1,2,5]. The importance of emotional behaviors, where emotion is identified as an essential and integrated entity, is not a recent idea. From Hippocrates to the Renaissance, the ethical-philosophical principle of mind–body dominance prevailed in medical techniques. The duality between the thinking mind and the physical body was significant in all areas of knowledge, and only in the 1990s did some scientists provide a foundation for psychoanalytic theory, proving that the existence of emotional behaviors is important for survival and reproduction, both in terms of cognitive function and the significance of primary emotions. Thus, we can observe that both the emotional mind and the rational mind intersect and complement each other, demonstrating that our emotions and how they are experienced influence our reasoning [8,9]. This phenomenon is crucial for understanding the user's experience during

a game, as it allows us to comprehend the choices and reactions, they had in a given situation [1,2,5].

According to various studies in the field of games, there is a consensus that emotions have a significant impact on digital games, especially in the following categories: (1) human–computer interactions; (2) effectiveness; (3) efficiency; and (4) player satisfaction [2,6,7]. One of the main motivating factors for player engagement in the experiences provided in games is the emotional experience [2–6]. Games that elicit emotion are crucial for a variety of purposes, including education, training, therapy, and other areas where the elicitation of emotions is vital [6].

### 1.2. Role of Virtual Reality

We can enjoy a pleasant experience when we interact with entertainment mechanisms through games [4]. Although there are many different game genres and each has unique qualities, games are often interactive, competitive, and goal-oriented [3]. Games ought to be dynamic, exciting, and enjoyable for the player by nature. Players tend to gravitate towards games that offer a pleasurable experience while simultaneously providing opportunities for adventure, challenge, or an adrenaline rush [3–5]. In these challenges, players are not necessarily looking for immediate and tangible rewards, as the emotional experience itself is impactful [5]. These emotional experiences are important for the user, whether they involve overcoming a difficult challenge in the game or being able to escape from real-world concerns [4].

Technology advancements have enabled games to be developed for a variety of platforms, including Virtual Reality. VR is considered a promising technology because it allows for immersion, interactivity, and presence, adding a better dimension to the world of games by making them as interactive as possible [1,7,10,11]. In order to enhance the player's experience, several game development companies have focused on improving physics, game graphics, and sound. Later on, narrative manipulation started to influence the player, leading to an improvement in the quality of characters and stories in the world of games. However, there is still a barrier between the player and the game, which is why emotions began to play an important role. It has been realized that any game can evoke a wide range of emotions, which can take the user experience to another level [2].

Since games have the ability to influence the player's emotional state, they can be used to elicit emotions in a natural and ethical manner. To evoke emotions, one can take advantage of images, sounds, music, and/or interactions [3,10,12]. What makes this elicitation of emotions more fascinating is the fact that players engage in the experience even if it involves negative emotions such as fear and frustration [5].

The way the player interacts with the game can be altered through manipulations of the user's emotional state. This can result in a much richer experience that deeply involves the player. To achieve different emotional states in a player, various techniques can be employed. To do this, attention must be given to three questions: (1) what the stimuli are, (2) when they should appear, and (3) how these stimuli adapt within the game [5].

Furthermore, players' preferences for game genres, as well as their experiences and personalities, affect their relationship with games. When developing a game with a focus on emotions, it is necessary to keep this in mind in order to achieve good design. Game design encompasses numerous elements, and it is important to determine which ones should be used to elicit an emotion. These elements can include non-playable characters (NPCs) and game content objects [5].

### 1.3. Role of Escape Room Game Genre

Escape Rooms are a genre of game focused on problem-solving. These problems usually involve puzzles that can be solved in various ways, such as through riddles or symbols. In addition, Escape Rooms are games that challenge players on a psychological level. The mind-body connection of Escape Room participants is pushed to the point where they genuinely believe they are a part of the narrative; in other words, the participants

truly take on the role of the character [13]. The objective of an Escape Room is to solve all the puzzles and uncover mysteries in order to "escape". This sense of "escape" can be interpreted in various ways, whether related to any concept or theme or in the literal sense of the player being physically trapped. By involving the player in a confined space filled with props and puzzles around them, the player's brain truly believes it needs to "escape" and enters survival mode. This phenomenon leads the player to be more attentive and focused on the tasks, primarily due to the surge of adrenaline. Escape Rooms are the genre of game that evokes the most emotions in a player, especially negative emotions, as they involve a dimension of gameplay elements such as triggers. These elements are typically used to provoke fear, anger, and frustration in the player [13–16].

Escape Rooms are designed to provide immersive experiences for players. To make the experience as engaging as possible, the participant must take on the role of a character in the game. This can be accomplished by making the player an active participant in a puzzle-filled environment. This releases adrenaline in the player's brain, making them more alert and focused on what they are doing, allowing for innovative thoughts. These thoughts can be useful for discovering clues or solving a problem. However, these thoughts can also be a limitation, as they may prevent the player from seeing an obvious answer because they assume it must be something complex [13].

Some types of puzzles can include: (1) riddles; (2) hidden objects; (3) mirrors; (4) UV light; (5) symbols; (6) crosswords; and (7) patterns, among others [14]. This type of game attracts many players simply because humans are naturally curious [16]. Participants need to be creative and explore the space to solve the proposed puzzles. Fear is an emotion commonly used in Escape Rooms, although it is not the only one—it is the most commonly used and has the longest-lasting effect [16].

Fear can be elicited in various ways, including the impression of being trapped or the fear of failure, both of which serve to challenge the player. Therefore, fear needs to be used as motivation to solve the puzzle as quickly as possible, arousing the player's creativity and attention. Another common feeling is enthusiasm, namely the desire for novel and enjoyable experiences. Even if the room is designed to induce fear or frustrate players, there are enjoyable aspects to counteract that. On top of that, it may be extremely satisfying to move forward and complete a goal, no matter how challenging or scary it may be. Sometimes the objective may not be as difficult, requiring the player to pay attention and be creative [16].

### 1.4. Objectives of the Study

This study, titled "Escape VR: The Guilt—An Escape Room in Virtual Reality for Eliciting Emotions," is driven by several key objectives aimed at comprehensively exploring the realm of Virtual Reality (VR) gaming and emotion elicitation. The primary objectives can be summarized as follows:

1. Game Development and Design: The foremost objective of this research is to meticulously document the game development process of "Escape VR: The Guilt". This includes an in-depth examination of the creative and technical aspects of crafting an immersive VR Escape Room experience designed to elicit a wide spectrum of emotions in players.
2. Challenges and Triumphs: To shed light on the multifaceted nature of VR game creation, this study aims to elucidate the challenges encountered during development. By offering insights into these challenges and how they were effectively surmounted, the research highlights the tenacity and problem-solving capabilities required for successful VR game design.
3. Emotion Elicitation: A central objective is to explore and analyze the game's capacity to elicit emotions in participants. The study seeks to identify, categorize, and quantify the emotional responses experienced by players while navigating the virtual Escape Room. By doing so, it strives to offer a comprehensive understanding of the emotional dynamics at play in VR gaming.

4.  Reflection on Results: Beyond the development process, this research seeks to reflect upon the outcomes obtained. Through careful analysis and interpretation of the elicited emotions, the study aims to discern patterns, trends, and player preferences. This reflective aspect contributes to the broader knowledge base concerning emotional engagement in VR gaming.
5.  Demonstrating Success: Ultimately, the study aims to showcase the success of "Escape VR: The Guilt" in achieving its intended goal of eliciting a diverse array of emotions in participants. By providing empirical evidence of emotional responses, the research underscores the game's effectiveness as a tool for emotional engagement in the realm of VR Escape Room experiences.

This paper is divided into five sections. The second section encompasses the literature review, where we will present two projects: (1) the first project shows what made players feel certain emotions, and (2) the second project is a Virtual Reality Escape Room game, where we can understand the precautions to be taken. The third section of this paper is where we explain all the processes of design and development related to the game, divided by: (1) objectives and goals; (2) platform and tools used; (3) prototyping and iteration; (4) game mechanics; (5) environments and scenarios; (6) interaction and controls; (7) emotions addressed; and (8) challenges and solutions in game development. Section 4 follows with the results, where we will present the evaluation of the first prototype in a Nursing Teaching Institution divided by: (1) the preparation; (2) testing; (3) questionnaire used; (4) questionnaire results; and (5) evaluation of results and feedback. In the last section, we will draw the conclusions of the project, presenting: (1) the main discoveries and contributions; (2) comparison to previous studies; and (3) future improvements and recommendations for future studies.

## 2. Emotion Elicitation Case Studies Analysis

### 2.1. XEODesign

XEODesign embarked on an extensive eleven-year investigation aimed at unraveling the intricate relationship between emotions and video games, with the ultimate goal of infusing future games with heightened emotional depth. Participants engaged in gaming sessions lasting between ninety and one hundred and twenty minutes within the comfort of their own homes. Their insights, thoughts, and emotional experiences were shared throughout these gaming sessions. Moreover, some sessions involved online group gameplay [4].

Through the analysis of more than 40 million player experiences and extensive research and design efforts, XEODesign sought to answer several fundamental questions [4]:

- The Spectrum of Emotions: The researchers delved into the myriad of emotions that players experience during gameplay. This included emotions like anger, excitement, and even moments of emotional vulnerability, such as crying;
- Emotions Beyond Story: XEODesign aimed to identify emotions in games that are not solely tied to the game's narrative but arise from the gameplay itself;
- Motivations for Play: They sought to understand whether players engage in gaming not just for the challenge but for the emotional experiences' games provide. This led to questions about where these emotions originate during gameplay;
- Emotion Modification: XEODesign investigated whether players actively modify their gameplay experiences to elicit specific emotions;
- Building Emotions into Gameplay: The study explored the possibility of integrating emotion-inducing elements or actions within gameplay, rather than relying on cutscenes. They also examined to what extent game developers were already implementing such strategies;
- Player Experience and Preferences: XEODesign uncovered that players engage with games not just for the games themselves, but for the unique experiences these games create. These experiences encompassed adrenaline rushes, immersive adventures, mental challenges, moments of solitude, or social interactions.

To address these inquiries, XEODesign conducted a rigorous research study involving 15 hardcore gamers, 15 casual gamers, and 15 non-players. The study encompassed diverse gaming environments, ranging from individual home settings to multiplayer sessions. Both qualitative and quantitative data collection methods were employed, including video recordings, questionnaires, and observations of facial and non-verbal cues during gameplay [4].

From this extensive research, XEODesign identified Four Keys to understanding how emotions manifest during gameplay [4]:

1.  Hard Fun: This key focuses on the emotional responses arising from meaningful challenges, strategies, and puzzles. Players often enjoy the thrill of overcoming obstacles, leading to emotions like Frustration and Fiero (a sense of personal triumph);
2.  Easy Fun: Here, the focus is on the sheer enjoyment of engaging in game activities. This key capitalizes on player curiosity, enticing them to explore game worlds and experience emotions like wonder, awe, and mystery;
3.  Altered States: This key revolves around the transformation of the player's internal emotional state during gameplay. Emotions like Excitement and Relief emerge through in-game interactions, thoughts, and behaviors;
4.  The People Factor: Emphasizing the social dimension of gaming, this key pertains to the emotions generated through player interactions, including competition, cooperation, and performance. It encompasses emotions like amusement, schadenfreude (pleasure from others' misfortune), and naches (pride in another's accomplishment).

In the fascinating landscape of gaming, the findings of this study unravel a profound truth: people engage with games not solely for the thrill of victory or the allure of a compelling storyline but to orchestrate, alter, and immerse themselves in a symphony of internal experiences [4]. According to this study's findings, adults use games as a means of building their emotional landscapes, finding refuge from daily tasks, experiencing the thrill of success, and exploring previously unexplored domains of wonder and curiosity [4]. Games also provide players with the freedom to design their own moments of relaxation, contentment, and self-affirmation, which adds to the fascinating kaleidoscope of emotions that motivate their gaming experiences [4]. In addition, understanding the "Four Keys" to unleashing emotions in video games provides valuable insights into the nuances of player experiences. These findings not only shed light on why we play games, but they also open up a world of opportunities for game designers to create emotionally powerful experiences by cleverly playing with these keys [4].

### 2.2. VR Escape Room Game

Samira and Layla, students from Al Yamamah University, created an Escape Room game to help players improve their skills. In this game, players solve puzzles to escape from different rooms. They also wanted to make sure players did not feel sick while playing in Virtual Reality (VR) [7]. To do this, they made it so players move using teleportation, which means they do not have to physically move in the real world. This VR Escape Room game is all about boosting problem-solving, thinking, and observation skills, among others [7].

The game mixes three types of experiences: horror, puzzles, and adventure, each in a different room. To move forward, players need to understand the stories in these rooms and solve puzzles involving math, patterns, and combinations. It is possible to go from room to room, and as it progresses, the challenges become tougher. Since players move with teleportation, the game is quicker, taking about 10 to 15 min per room [7].

The game starts with a menu, and once It begins, puzzles need to be solved. Each room gives a limited number of tries and if the player fails too many times, the game ends. They have four, three, and one tries for the first, second, and third rooms, respectively. If the player can use their skills and finish all the puzzles within these attempts, they win the game [7].

To create this Escape Room game, they used the HTC Vive Virtual Reality headset [17] Unity3D [18] as the game engine with the C# programming language [19], VRTK [20], and

SteamVR [21] for VR support, and Blender [22] for making 3D models [7]. They designed the game's movement to reduce motion sickness, which can happen when the actions taken in VR do not match the real world. Besides teleportation, the game moves based on where the player looks, which makes it feel more natural [7].

When it starts, the player picks "start game" to get to the first room. To open the door to the next room, the player needs to touch the door with the right answer. If it is right, it turns green. Math puzzles need a key as the answer, while pattern puzzles use a cube, as shown in Figure 1 [7].

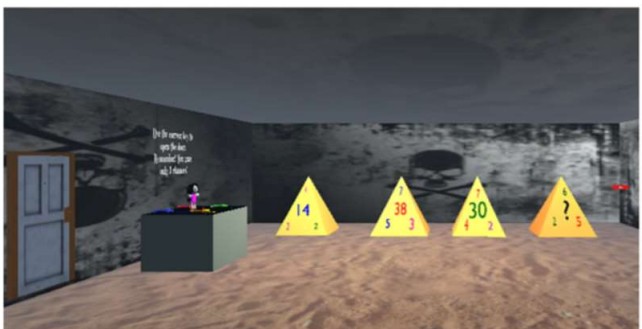 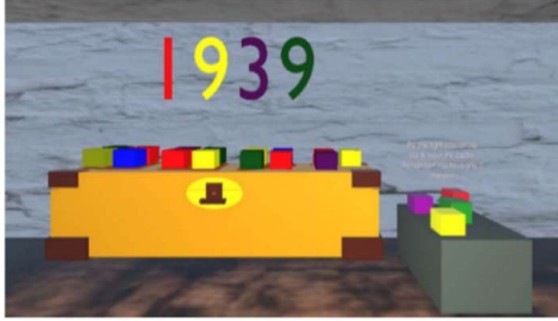

**Figure 1.** VR Escape Room game puzzle examples.

As the game was developed to be, among other things, a horror game, the environment is scary and the background music adds to the experience. They also created a menu where you can learn about each room's story [7].

For testing, they split participants into two groups: one without teleportation (they had to move physically) and one with teleportation (they used the HTC Vive controllers to move). Figure 2 shows how much time each group took to finish the game and how sick they felt. The time is measured in minutes [7]. From Figure 2, we see that the group without teleportation did not take a lot more time to finish the game, but they felt sicker. The shorter time in the game helped with motion sickness, but people who took more time felt more immersed [7].

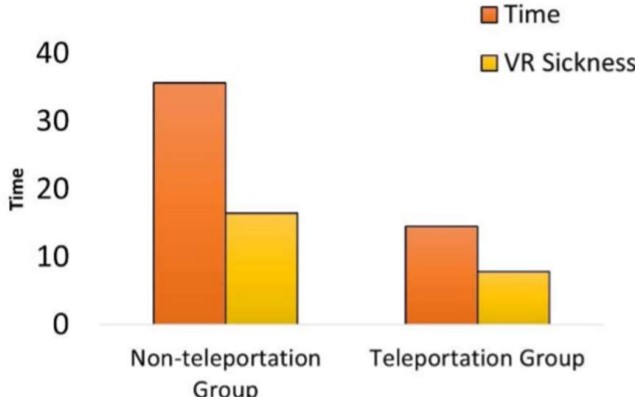

**Figure 2.** Relationship between time and VR sickness level felt by non-teleportation and teleportation groups [7].

This VR Escape Room game, as designed and implemented in this research, serves as an intriguing case study in the realm of emotion elicitation through Virtual Reality experiences. The study effectively taps into the immersive potential of VR to elicit a range of emotions in players. By combining horror, puzzle-solving, and adventure elements, the game creates an environment that naturally triggers negative emotional responses. Horror, for instance, can evoke fear, tension, and suspense, while challenging puzzles stimulate frustration, curiosity, and satisfaction upon their completion. In contrast, adventure elements

often introduce excitement and anticipation. One of the notable findings is the correlation between the time spent in the VR game and the depth of emotional engagement. Players who dedicated more time to the game reported a stronger sense of immersion. This is a significant insight because it indicates that VR experiences have the potential to establish profound emotional connections with users, akin to traditional forms of entertainment like movies or literature. This emotional depth can contribute to a more memorable and impactful experience.

In conclusion, this study showcases the multifaceted potential of VR technology in eliciting emotions and enhancing the overall user experience. It not only highlights the effectiveness of this VR Escape Room game in stimulating emotions but also underscores the importance of user comfort in sustaining engagement.

### 3. Game Design and Development

#### 3.1. Objectives and Hypotheses

The objective of this game was to develop an Escape Room where, while the players explore and complete puzzles to progress in the game, triggers would be added to elicit positive and negative emotions through visuals or sounds, as well as using the puzzles themselves. Having this in mind, we needed to be careful while triggering negative emotions, as we wanted to keep players engaged and motivated to complete the game.

With this, we created seven different hypotheses that lined up with the game objectives as well as the themes explored with this project: (1) emotions; (2) Virtual Reality; and (3) elicitation of emotions in games. All these hypotheses will be explored in the Conclusion section of this paper. The seven hypotheses are as follows:

**H1.** *It is not possible to predict the exact emotions that will be felt by the players;*

**H2.** *The players are not able to identify everything they feel;*

**H3.** *Virtual Reality contributes to a better player experience;*

**H4.** *Eliciting negative emotions does not make the players give up on the game;*

**H5.** *Eliciting negative emotions does not make the players dislike the game;*

**H6.** *Negative emotions affect the puzzle-solving time;*

**H7.** *Different individuals can experience multiple emotions, both positive and negative, in response to the same game element.*

#### 3.2. Game Specification

The game "Escape VR: The Guilt" is a Virtual Reality game that falls under the genres of Escape Room and mystery. The player finds themselves in a dark house, where they must complete various puzzles in a linear fashion, meaning each puzzle unlocks the next one. Throughout the game, the player will come across several clues that unravel parts of the narrative. This narrative focuses on the lives of the game's characters, John and Mary, and although there is a storyline, it remains open-ended, allowing each player to have their own personal interpretation of what happened. In order to make the puzzles more challenging, the player is immersed in this dark environment filled with elements designed to stimulate the player's emotions.

As mentioned before, the player will have access to clues that reveal part of the narrative; these clues are: (1) the pages of the diary (Figure 3); and (2) post-its (Figure 4).

Each page of the diary features a date, followed by a text written by John, and ends with his signature. In total, the game contains six pages that the player can find in the following rooms: (1) one page in the living room, (2) two pages in the bedroom, and (3) three pages in the office. The objective of this game element is to convey the narrative to the player in a way that is engaging and arouses curiosity about what happened. The open-ended nature of the narrative, especially in the diary pages, is intended to pique the

player's curiosity, allowing them to form their own perception of John and Mary's story. The player gradually obtains the pages as they progress through the game, but it is only in the office that they discover the need to connect the diary pages in a specific order to obtain a code. This code is used to lock a cabinet in the laboratory. Upon successfully opening it, the player sees their reflection in the mirror and realizes they are John.

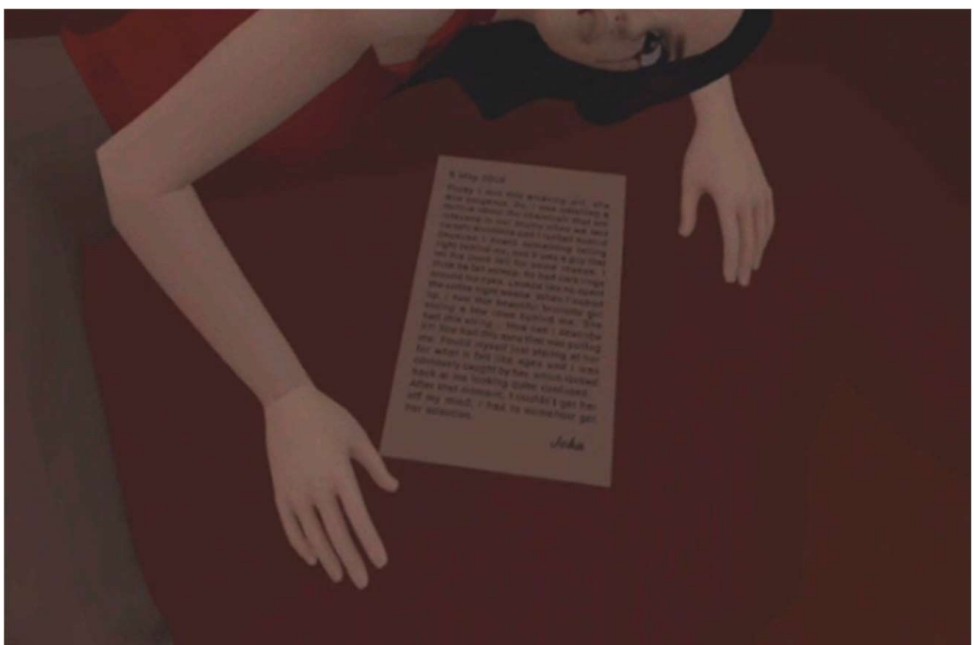

**Figure 3.** Example of pages of the diary.

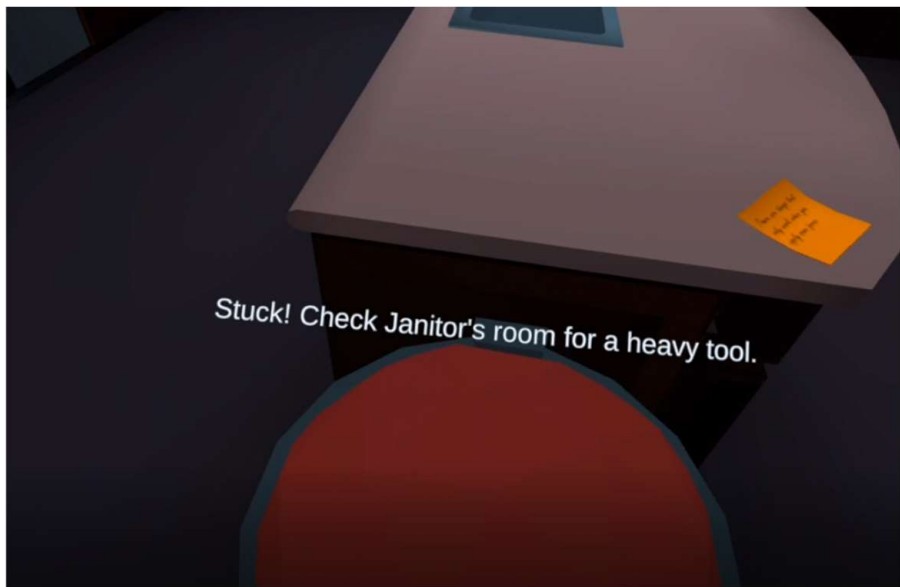

**Figure 4.** Example of post-its.

The post-its are presented near the object that players need to interact with, in order to help them find their way. As players find the object and try to interact, a new clue will instruct them on what to do and what tools to use, as Figure 4 shows.

### 3.3. Emotions Addressed

Based on the research gathered in the state-of-the-art, we can verify that it is not possible to guarantee that a player will feel exactly the emotion we have in mind. However,

using Norman's generic list of positive and negative emotions, we can predict whether a certain element will evoke a positive or negative emotion [1,2,23,24]. Therefore, since the game aims at Eliciting Emotions, we created, in collaboration with our partners, a list of elements that we wanted to serve as triggers for positive and negative emotions. These elements were based on the research findings from XEODesign [4]. In Table 1, we can see the elements that were designed as triggers for positive emotions, and, in Table 2, we can see the elements that were designed as triggers for negative emotions.

**Table 1.** Positive emotions trigger elements.

| Element | Description |
| --- | --- |
| Completing Objectives | The player is able to complete objectives after several attempts. |
| Progressing in the Game | The feeling of game progression when completing an objective. |
| Game Mechanics | Because they are simple to learn and interesting to use. |

**Table 2.** Negative emotions trigger elements.

| Element | Description |
| --- | --- |
| Narrative and Environment | The narrative and game environment provide the story of John and Mary, where the player gradually discovers what happened. The player has an open-ended story to fill in the gaps with their own interpretation. |
| Easy-to-Solve Puzzles | Leading the player to believe that the puzzle will be complicated, while in reality, it is quite simple. |
| No Inventory | This way, the player can only carry one item in each hand. This requires them to move around the house more often. |
| Dark Environment | The game environment is dark, with some parts completely dark that can only be navigated with the flashlight. |
| Flashlight | The flashlight does not automatically point in the direction the player is looking. The player must manipulate the controller to aim the flashlight where they want. Additionally, the player needs to be careful not to lose the flashlight in the dark. |
| Cockroaches | Cockroaches appear in the trash, where the player has to reach into them to retrieve a key. |
| Chest Lock | It requires the player to crouch down and be at the correct distance to rotate it to the correct numbers. |
| Mary's Body and Eyes | Mary's body changes position every time the player looks away. When the player approaches the body, Mary's eyes follow the player. |
| Sounds | Using three whispers from a woman when the player is on the stairs, in the dark. |
| Heights | The player has to pass through a hole in the balcony of the room using a narrow plank. |

*3.4. Game Mechanics*

3.4.1. Locomotion

The possibilities for locomotion are countless in a virtual environment. However, in order to guarantee continued immersion, it was crucial for us to ensure that the movement was as authentic as possible. We started with a slightly different locomotion system, in which the players only had to move their hands forward and backward, as if they were walking in real life, to move in the direction they were looking. However, once we began interacting with various game elements, small movements required the user to stop what they were doing in order to move slightly, which ended up completely breaking

immersion. Following this insight, we were divided between incorporating joysticks or teleportation. This is due to the fact that, on the one hand, joysticks can provide a more immersive experience while also increasing the risk of motion sickness, and, on the other hand, teleportation has a low risk of motion sickness but greatly disrupts immersion in our game. We decided to proceed with joysticks, using the input from the left controller's joystick to move forward, backward, and sideways, and the right controller's joystick for 45-degree rotations that help facilitate movement within the environment.

### 3.4.2. Grabbables

These are all the objects that the player can pick up during the game. Throughout the game planning, we realized what all the objects the player can pick up have in common and what they need to function: picking up, dropping, simulating physics, and occlusion. All grabbable objects have a green outline. This way, the players know the important objects in the game and do not waste time looking for them. Figure 5 shows an example of a grabbable.

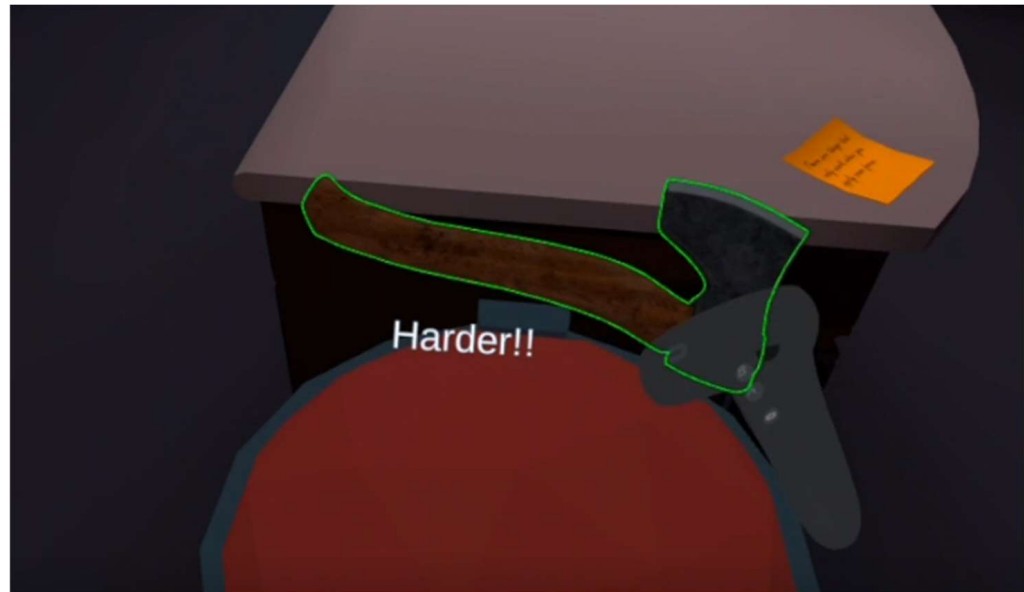

**Figure 5.** Example of a grabbable.

### 3.4.3. Objectives System

This system controls the appearance of objectives to guide the players on what they need to do. At the beginning of the game, the first objective for the first puzzle to be completed appears for the first time. Subsequent objectives only appear sequentially after completing the previous puzzle. Figure 6 shows an example of an objective.

### 3.4.4. Doors

For the players to be able to enter a room, they need to insert the key corresponding to that door to be able to unlock it. After that, the players need to use the commands to open the handle as a grabbable and push the door open. If the player inserts the wrong key, it will fall to the ground, and a warning will appear saying that the key is not the correct one.

### 3.4.5. Trash Puzzle

This is the first puzzle that players will interact with. In order to complete this puzzle, they need to beat the trash lid with a hammer using a specific amount of force. If the players apply less force, it will tell them to hit harder. If they use the wrong tool, it will warn them. When the players completely open the lid, it will show a key to the next door, and the cockroaches are triggered to animate around the trash. Figure 7 shows the trash opened.

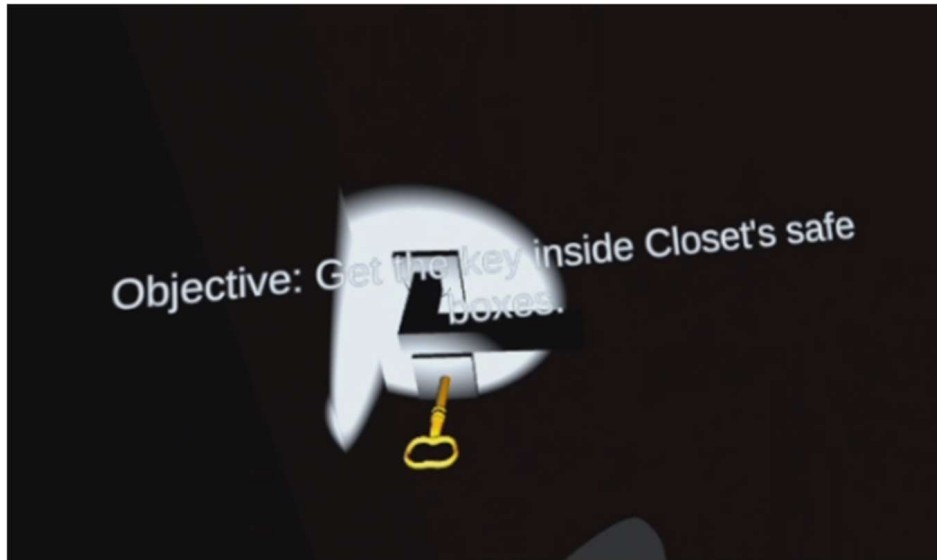

**Figure 6.** Example of an objective.

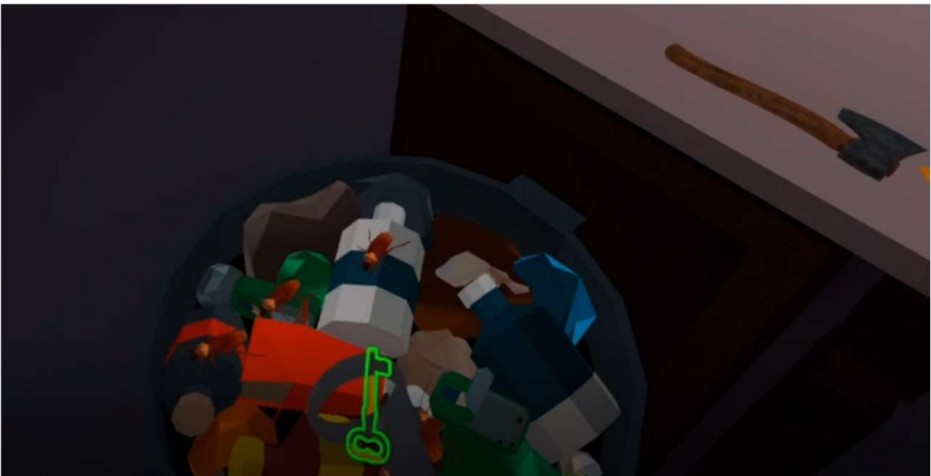

**Figure 7.** Trash opened with grabbable key and cockroaches.

3.4.6. Chest Puzzle

In this second puzzle, the player's objective is to decipher the code for the chest lock, which consists of four digits, in order to obtain the key to another room in the house and the flashlight that allows them to navigate in the dark. We decided that the code would be the month and day of a specific date. As a clue for the player, we scattered various frames with dates throughout this room, where the chest is located. We also placed the first page of one of the diary pages near the chest. All the diary pages have a date, and the idea is for the player to notice that the same date on the diary page is also present on one of the frames. We specifically placed this frame as close to the chest as possible. To enter the code on the chest lock, the player simply needs to crouch down and use the Hand Trigger button, which is also used to grab grabbables, to click next to the number they want to modify. The number will then rotate in the positive direction (if it is on one, it moves to two, if it is on two, it moves to three, and so on until it reaches nine, which then moves to zero). Figure 8 shows the chest puzzle.

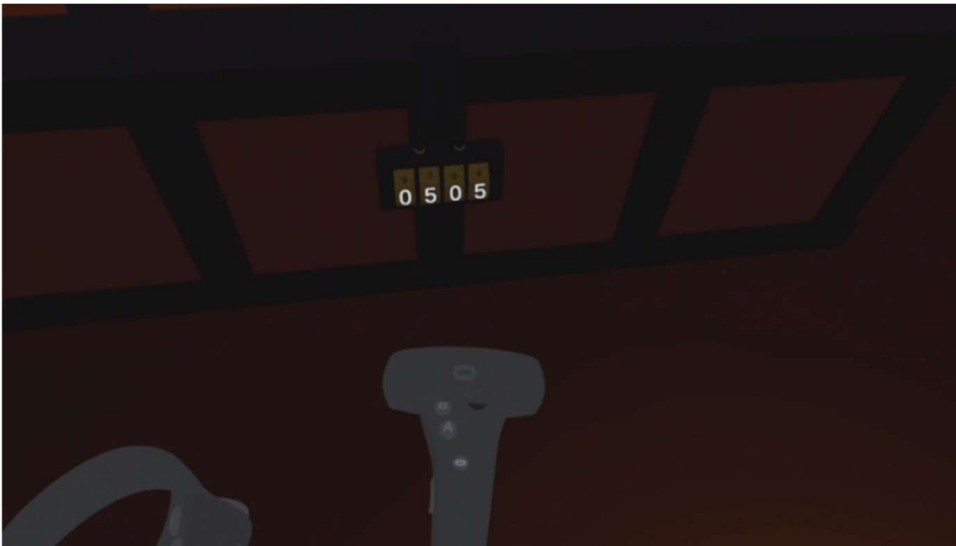

**Figure 8.** The chest puzzle.

### 3.4.7. Flashlight

The flashlight is a grabbable object, which means that when the player picks it up and releases it, it turns the flashlight on and off, respectively. For this flashlight, we wanted a realistic behavior where, when we point it towards a dark area, that area becomes illuminated, taking into account the distance from the flashlight. The player finds this flashlight after completing the chest puzzle and can use it in the dark areas of the house to see the path and objects.

### 3.4.8. Safes Puzzle

In this puzzle, the player's objective is to find the keys scattered around the Bedroom area, the same area where the three safes are located, in order to unlock the safes and find the key to the next area, which is only in one of them. The function of the keys is the same as that of the doors. Figure 9 shows the chest puzzle, with the player using the flashlight to be able to see.

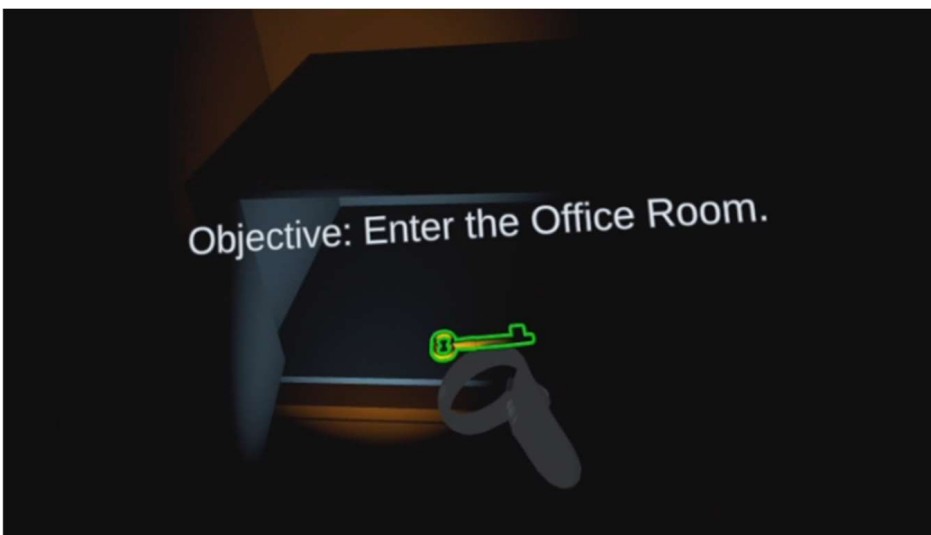

**Figure 9.** The chest puzzle with flashlight.

### 3.4.9. Diary Pages Puzzle

Upon completing this puzzle, the player unveils the final word that will be needed for the last puzzle. Here, the player must collect all six diary pages and place them in the correct order, revealing the letters as they are placed on the board. The positioning of the diary pages on the board must be from left to right. If the position where the player is trying to place the page is already occupied, the page simply falls down. Otherwise, we programmatically animate the movement and rotation of the page to the position on the board and reveal the correct letter, displaying it beneath the page. Once the player places all the pages, if they are not in the correct position, a message is displayed indicating that the positioning of the pages is incorrect. Otherwise, the puzzle is considered completed, and a message is shown to the player indicating the next objective, which is a clue for the next puzzle, the bookshelf puzzle. It activates the television in the same room (Office), which serves as a clue with three symbols for the next puzzle, the bookshelf puzzle. Figure 10 represents the diary pages puzzle, and Figure 11 is the television with the clue for the next puzzle.

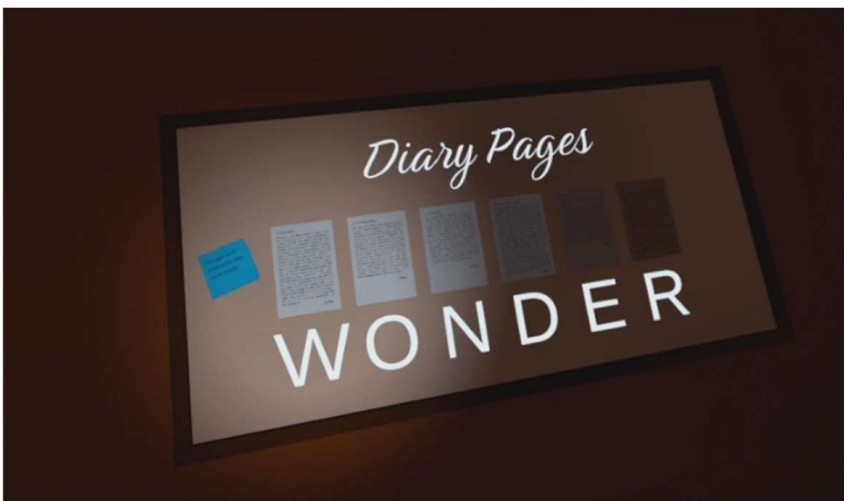

**Figure 10.** The diary pages puzzle.

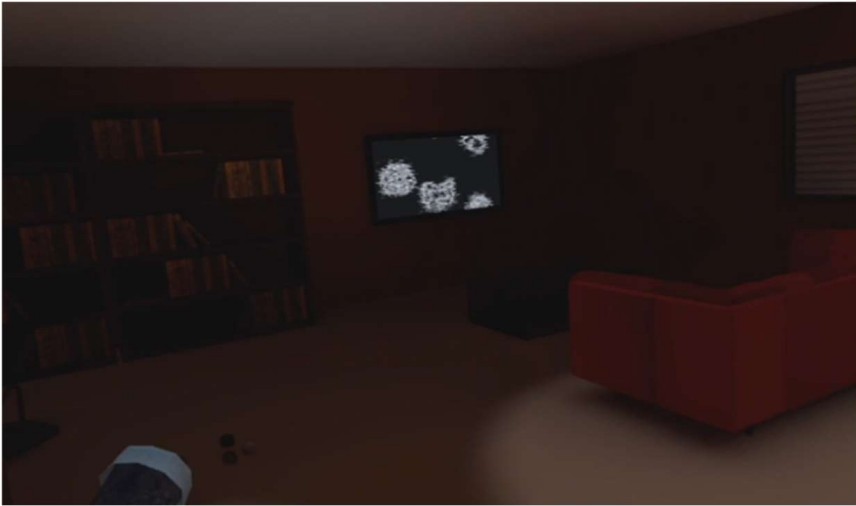

**Figure 11.** TV with clues.

One of the pages required to complete this puzzle is on the balcony, which itself is a small puzzle. Here, the player must realize that they need to use a wooden plank (grabbable) that is on the balcony and place it over the hole in the balcony that prevents the player from obtaining the diary page. Figure 12 shows the wooden plank in place to allow the player to get the diary page.

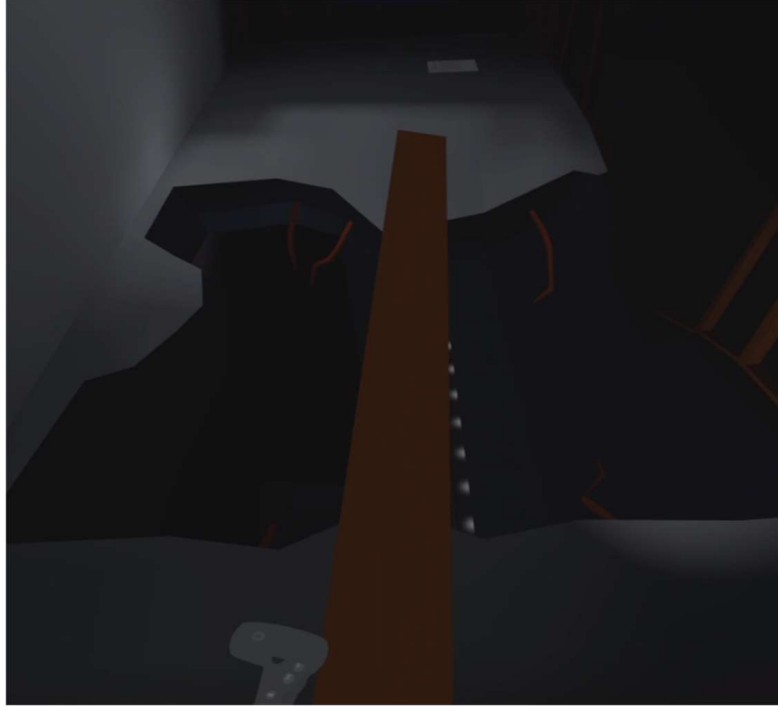

**Figure 12.** Wooden plank.

### 3.4.10. Bookshelf Puzzle

In this puzzle, the player must find the correct books among all those on the bookshelf in order to open it and access the laboratory. The previously stated television provides the hint for the correct books, displaying a total of three symbols that are marked on the spines of the correct books. All the books have symbols on their spines, but only three of them correspond to the symbols shown on the television. Once the player finds the books, they simply need to grab them, and the books will animate to protrude further from the shelf. When all three books are activated, a secret door opens to the next room.

### 3.4.11. Mirror Puzzle

In this puzzle, the player must use the word revealed in the diary pages puzzle to open the small cabinet in the laboratory. The player needs to put small-lettered pieces that form the correct word in the correct order, through the cabinet's lock. These pieces can be found scattered around the laboratory. Once the player completes the puzzle, the lock animates to fall, and the cabinet door opens. Inside, the player encounters a mirror where they realize they are John. Figure 13a shows the locker, and Figure 13b shows John in the mirror.

### 3.5. Controls

The controls for this game are very simple to learn and use (Figure 14). In order to move through the environment, the player needs to use the joystick (1) in the left hand. To be able to move the direction by 45 degrees, the player needs to use the joystick (1) in the right hand. To be able to grab the objects, the player needs to use the back button (5) of either command. Figure 14 shows the controls of an Oculus Quest command. On the right hand, we can also find two buttons with the letters "B" and "A". In order for the players to

see the present objective, if they forget, they can press the "A" button, and the message will appear again.

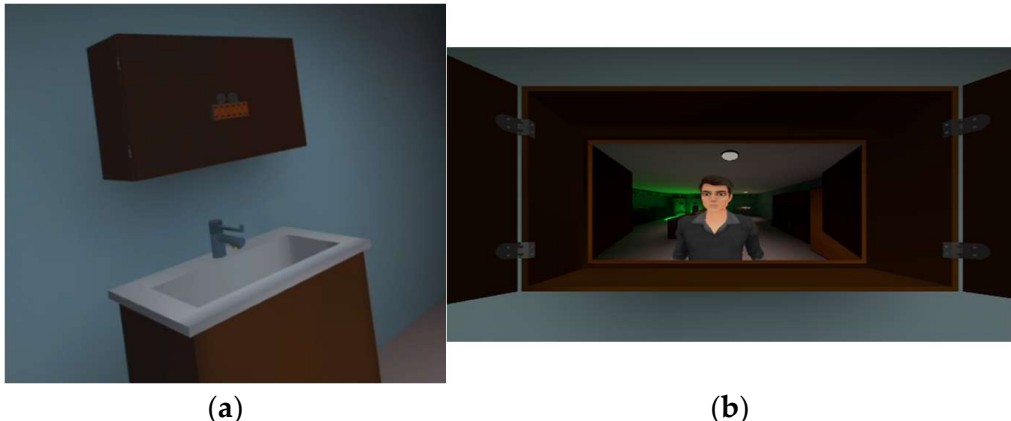

(**a**)                 (**b**)

**Figure 13.** (**a**) Locker; (**b**) player in the mirror.

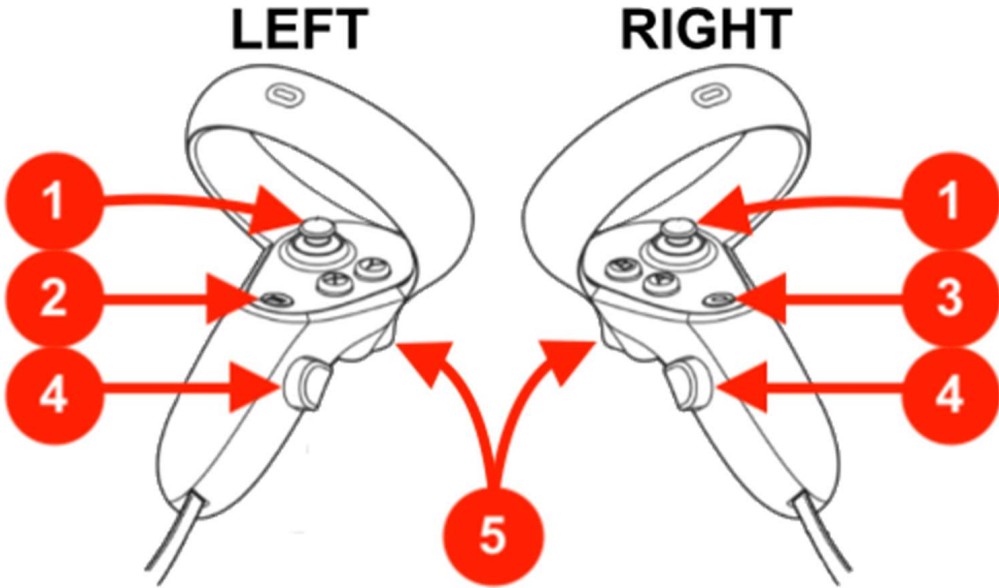

**Figure 14.** Game controls (1—thumbsticks (joysticks); 2—menu button; 3—oculus button; 4—grip buttons; 5—triggers/grip (back buttons).

*3.6. Challenges and Solutions in Game Development*

3.6.1. Movement through Objects

Virtual Reality always poses a significant challenge in terms of gameplay design since it relies on the player's real-world movement, which cannot be controlled. With that in mind, we had to come up with a solution for the issue of the player passing through objects they should not be able to, such as walls or doors. The challenge here was not when using the joystick, as that can be easily restricted with a simple check: whether the next position will or will not pass through the wall. The real challenge lay in the player's physical movement within the game space. We had two initial solutions. One was to darken the screen until it became completely black when the player tried to pass through any object we did not want them to see until the appropriate moment, and then block their movement in the game, preventing the player from feeling motion sickness when the game stops moving with them. The second option involved pushing the player backward whenever they tried to pass through a collider. The results of this latter solution were much more realistic than the former, so we proceeded with the second option.

### 3.6.2. Grabbables

After the first internal test, we realized that most players had difficulty understanding what a grabbable object was, i.e., an object that could potentially be important for solving a puzzle, and what was not. To address this problem, we decided to add an outline around objects, which we easily achieved using a plugin we found in the Unity Asset Store called Quick Outline. We created a simple check within the Update event that uses the previously mentioned Overlap Sphere method to detect hands within a certain distance. If a hand is detected, the outline is turned on; otherwise, it remains off.

Shortly after the first internal test, we encountered a problem that required a quick solution. It was possible to grab a grabbable object and pass it through other objects, such as a wall (in which case it would fall on the other side of the wall when released). To address this situation, we first had to create a method that checked whether the object was being rendered by the camera or not. Initially, we used the "isVisible" variable exposed in the Unity Mesh Renderer class to check if the object was within the camera's field of view or not. However, this variable did not provide reliable results. Therefore, we implemented our own solution, where we perform a LineCast (similar to a Raycast but with a start and end point instead of just a start point and direction) for each vertex belonging to the Mesh Renderer's Bounding Box (in case there are multiple Mesh Renderers, LineCasts are calculated for all Bounding Boxes). If at least one of the LineCasts does not detect any collisions with objects in the scene, it means the object is not completely occluded by another object. In such cases, when the object is released, it simply falls without any restriction. Otherwise, a Raycast is performed from the player's head towards the center of the grabbable object, and when the ray hits another object, the grabbable object is moved to that point. Then, a force is applied to the grabbable object in the direction of the player's face to ensure it does not get stuck inside the other object.

### 3.6.3. Flashlight

We created the lantern mesh in Blender, and in Unity, we added the Light component in Spot Light mode to the front part of the lantern. We realized that the Unity Standard Renderer Pipeline was somewhat limited for our purposes, as the light effect appeared unrealistic due to the baked shadows not being completely eliminated when the lantern received light. For this reason, we decided to switch to the Universal Renderer Pipeline, which not only provided the desired light effect with less effort but also made all colors more saturated, resulting in a more appealing environment. To control the lantern, we created a new class that contains the necessary methods to turn the light on and off. Since the lantern is a grabbable object, we used the events (mentioned earlier) that are triggered when the player picks up and releases the lantern as input triggers to turn the light on and off, respectively. Initially, we wanted the lantern to always be on. However, due to our limited time, we could not find a better solution to the problem of the lantern's light passing through objects like walls. We also attempted to control the lighting distance using the Raycast method to determine the distance to the objects the lantern is pointing at, but the results were not satisfactory.

### 3.6.4. Mirror

Initially, we wanted a realistic mirror effect where the player's movement and head rotation were calculated to create a more realistic effect. Additionally, we wanted to use Inverse Kinematics on the upper limbs to give the player the perception that they were seeing themselves in the mirror. We ended up adding Inverse Kinematics to the character's hands using a plugin available in the Unity Asset Store called Fast IK. However, we realized that making the mirror effect more realistic was not necessary because, at the moment the player completes the puzzle, their movement is blocked, and after just seven seconds, we darken the screen until it becomes completely black, with the only thing visible being a final message thanking the player for playing.

### 3.6.5. Illumination

Lighting is one of the most important and challenging aspects of this game. We used Unity's baking system to avoid dynamic lighting for performance reasons. Due to the complexity of our environment, the results were not as strong as we would have liked, but they were still considered decent. Since we did not have dynamic lighting, props that were not static would not be influenced by ambient light. To solve this problem, we added Light Probes. By carefully placing these probes throughout the scenes and performing a light bake, each probe retains the lighting information at that point and thus influences dynamic objects that pass through those points.

### 3.6.6. Optimization

Optimization is a crucial part of Virtual Reality development, especially on the Oculus Quest, as it has limited processing power, which is the trade-off for its incredible freedom as a standalone device. The first and most important indicator of how optimized the game is, is the frames per second (FPS), which needs to be consistently at 72 FPS (as the Oculus Quest limits the FPS to 72). Even occasional FPS drops can make players feel nauseous, so it is important to avoid them as much as possible, although it is not an easy task. Throughout the development process, we were very careful about the amount of processing power we were using for all the necessary calculations, such as physics and distance calculations, which were constantly needed. This was obviously important for optimization, but we quickly realized that our main issue was with the visual aspect.

A game can be CPU-bound, meaning that the performance impact is mostly related to non-visual calculations. However, in our case, the worst scenario was that the major performance impact was related to visual calculations (GPU-bound), and ideally, we wanted a balance between the two. It was the worst of both possibilities because minimizing visual processing is not an easy task.

Initially, we wanted to create a highly realistic environment, full of details. However, we reached a point where we had to find another solution because it was impossible to maintain the number of Batches and SetPass calls within the necessary limits to keep the FPS consistently at 72. Batches refer to the number of objects that the CPU can combine if they have the same material and are static, while SetPass calls are related to the need to switch to another material. For example, if we have two objects with the same material and they are static, we have one Batch and one SetPass call. If they are not static, we have two Batches and one SetPass call. If we have two different materials, even if they are static, we have two Batches and two SetPass calls. This gives us an idea of how limited we became.

Therefore, we decided to change our visual style to a more cartoonish style, taking advantage of a single material with a lightweight shader and a texture that we used to texture our models. Some of the props we obtained from the Asset Store retained their own materials but, since there were only a few, we did not consider it a significant issue. With careful consideration to make everything that could be static actually static and using baked lighting (removing all dynamic lighting except for the lantern), we were able to achieve the desired 72 FPS for almost the entire game.

## 4. Results

### *4.1. Prototype Testing*

#### 4.1.1. Preparation

At the beginning, participants were informed that this game was an Escape Room designed to elicit emotions. We explained the basic concept of the game, emphasizing that each objective would be clear and that they should focus on that to minimize exploration and speed up testing. Finally, we provided some tips, specifically for participants who were experiencing VR for the first time, on how to properly adjust the Oculus Quest to their face to achieve the best possible image quality. Once the participant indicated that they were satisfied with the adjustment, we instructed them to press the A button to start the game.

### 4.1.2. Participants

The tests for this project were conducted with two groups of people. The first group consisted of five individuals from the nursing field, and the other five individuals were from the gaming field. With a total of ten participants, the objective of these two groups was to evaluate the game with the initial target audience of our partners, but also to include more experienced participants to assess the differences between the two groups. The use of individuals from such diverse areas with different experiences in gaming and Virtual Reality was very interesting in understanding the gameplay process. This way, we could determine if the game is suitable for any user through an analysis of data from various player types. It is important to understand the difficulties and suggestions in order to improve this prototype in the future.

The tests with the nursing group were conducted at the University of Minho, Portugal. These tests started at 9:30 AM and ended at 2:00 PM. All participants in this group were female, aged between 21 and 30 years. The overall gaming experience level of the participants was 3. Out of the five participants, only one mentioned having previous experience with Virtual Reality, but she was always accompanied. Only one participant was able to complete the game, as the others started feeling nauseous and were instructed to end the experience. One disadvantage of this group was the lack of English language proficiency. Since the game is entirely in English, four out of the five participants required external assistance to translate the on-screen text. Table 3 shows these sociodemographic elements of the first group, including their profession or field of study, overall gaming experience level, previous experience with Virtual Reality, and the approximate time each participant took to complete (or not) the game "Escape VR: The Guilt".

**Table 3.** Group 1 sociodemographic description.

|   | Genre | Age | Area | Level | VR | Time |
|---|-------|-----|------|-------|-----|------|
| 1 | Female | 23 | Nursing | 2 | No | 1 h 15 min Finished |
| 2 | Female | 30 | Nursing | 4 | Yes | 45 min Not finished |
| 3 | Female | 21 | Nursing | 3 | No | 55 min Not finished |
| 4 | Female | 23 | Nursing | 3 | No | 50 min Not finished |
| 5 | Female | 23 | Nursing | 3 | No | 45 min Not finished |

The tests with the second group, consisting of participants from the gaming field, were conducted at the Polytechnic Institute of Cávado and Ave, Portugal. These tests took place throughout the remaining day until 9:00 PM. Two participants in this group were female, while the other three were male. The ages of these participants ranged from 23 to 42 years. Despite all participants being from the gaming field, they had different experiences: one participant had experience in game design, two participants had experience as artists, with one of them being a professor in the field, and the remaining two participants were experienced in game development. All participants had a high level of overall gaming experience. Only one participant had never experienced Virtual Reality before. Only the first participant in this group was unable to complete the game, unfortunately, because the Oculus Quest devices ran out of battery and the participant was unable to wait for them to recharge. Similar to the previous table, Table 4 represents the sociodemographic elements of the next five participants, including gender, age, profession or field of study, overall gaming experience level, previous experience with Virtual Reality, and the approximate time it took them to complete (or not) the game "Escape VR: The Guilt".

**Table 4.** Group 2 sociodemographic description.

|   | Genre | Age | Area | Level | VR | Time |
|---|---|---|---|---|---|---|
| 1 | Female | 25 | Game Design | 5 | No | 45 min Not finished |
| 2 | Male | 42 | Artist and Professor | 5 | Yes | 20 min Finished |
| 3 | Male | 27 | Game Developer | 5 | Yes | 20 min Finished |
| 4 | Female | 23 | Game Developer | 4 | Yes | 15 min Finished |
| 5 | Male | 23 | Artist | 4 | Yes | 50 min Finished |

*4.2. Prototype Evaluation*

4.2.1. Questionnaire Results

After each participant completed their experience, they were asked to move to another room to answer the questionnaire. Someone was present to monitor the responses and assist with any questions related to the questionnaire. All participants answered the questionnaire to the best of their ability.

For each section of the questionnaire, we will present a calculation for a 95% confidence interval for the mean values of emotions experienced in both the Nursing and Games groups, considering that each group had five participants. To do this, we considered the use of the standard error of the mean (SEM) formula: *SEM = Standard Deviation/√(Number of Participants)*. Then, we considered the use of the SEM to calculate the margin of error (MOE) and construct the confidence intervals: *Margin of Error (MOE) = SEM \* Critical T-Score (for a 95% confidence level with 4 degrees of freedom)*. Finally, we calculated the confidence intervals: *Confidence Interval (CI) = Mean ± MOE*.

Emotions Experienced

In the first section of the questionnaire, players selected the emotions they felt during the game. A total of fifty-one emotions were reported by the ten participants (Figure 15).

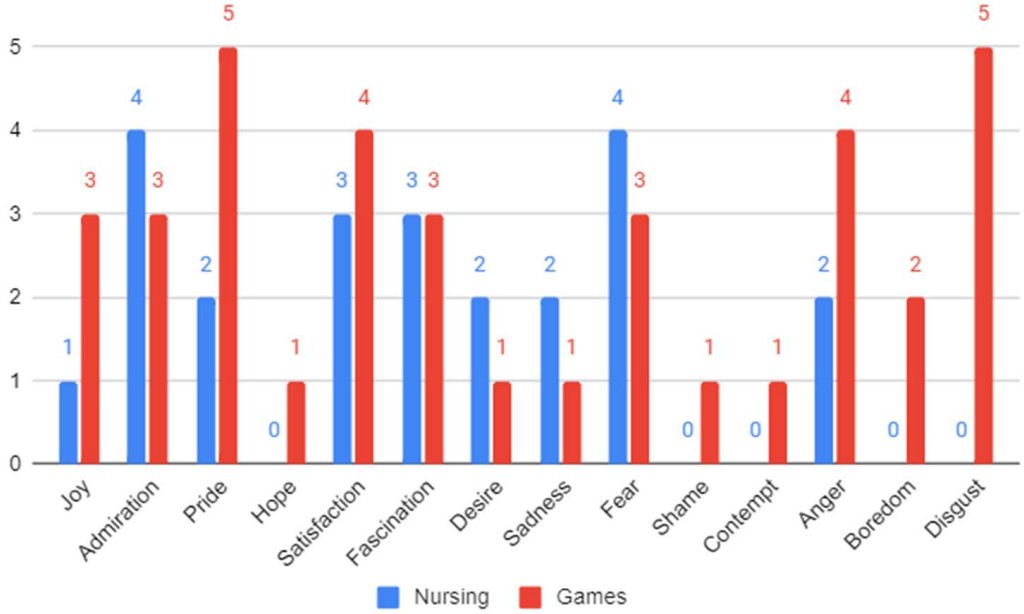

**Figure 15. Section** 1 questionnaire results.

Admiration, Pride, Satisfaction, Fear, and Disgust were the most frequently experienced emotions. Conversely, emotions like Hope, Shame, and Contempt were seldom felt, with only one participant experiencing each. The table below (Table 5) depicts the emotions felt by all participants, differentiating between the Nursing and Gaming domains, and shows the results of their confidence intervals.

**Table 5.** Standard deviation (SD) and 95% confidence interval (CI) calculated for each emotion felt by each group.

| Emotion | Group | Mean | SD | 95% CI |
|---|---|---|---|---|
| Joy | Nursing | 1 | 0.447 | 1 ± 0.350 |
| | Games | 3 | 0.488 | 3 ± 0.380 |
| Admiration | Nursing | 4 | 0.894 | 4 ± 0.699 |
| | Games | 3 | 0.577 | 3 ± 0.451 |
| Pride | Nursing | 2 | 1.140 | 2 ± 0.891 |
| | Games | 5 | 0.577 | 5 ± 0.451 |
| Hope | Nursing | 0 | 0.000 | 0 ± 0.000 |
| | Games | 1 | 0.577 | 1 ± 0.451 |
| Satisfaction | Nursing | 3 | 1.140 | 3 ± 0.891 |
| | Games | 4 | 0.894 | 4 ± 0.699 |
| Fascination | Nursing | 3 | 1.140 | 3 ± 0.891 |
| | Games | 3 | 0.577 | 3 ± 0.451 |
| Desire | Nursing | 2 | 0.894 | 2 ± 0.699 |
| | Games | 1 | 0.577 | 1 ± 0.451 |
| Sadness | Nursing | 2 | 1.140 | 2 ± 0.891 |
| | Games | 1 | 0.577 | 1 ± 0.451 |
| Fear | Nursing | 4 | 0.894 | 4 ± 0.699 |
| | Games | 3 | 0.577 | 3 ± 0.451 |
| Shame | Nursing | 0 | 0.000 | 0 ± 0.000 |
| | Games | 1 | 0.577 | 1 ± 0.451 |
| Contempt | Nursing | 0 | 0.000 | 0 ± 0.000 |
| | Games | 1 | 0.577 | 1 ± 0.451 |
| Anger | Nursing | 2 | 1.140 | 2 ± 0.891 |
| | Games | 4 | 0.894 | 4 ± 0.699 |
| Boredom | Nursing | 0 | 0.000 | 0 ± 0.000 |
| | Games | 2 | 0.577 | 2 ± 0.451 |
| Disgust | Nursing | 0 | 0.000 | 0 ± 0.000 |
| | Games | 5 | 0.894 | 5 ± 0.699 |

Triggers for Emotions

In the second section, participants described the game elements that evoked these emotions. Here are the emotions experienced within the Nursing group, along with their associated triggers, as provided by the participants:

- **Joy:** "The act of playing brought me joy".
- **Admiration:** "When I found John's body"; "The fact that the game takes place in a house"; "I admired the graphics, the arrangement of objects, how they were scat-

tered throughout the house"; "I was amazed to see the cockroaches because I didn't expect them".

- **Pride:** "I felt proud to be able to complete the challenges"; "I felt proud when I could complete the challenges".
- **Satisfaction:** "Solving the puzzles in the diary and the laboratory"; "I felt satisfied for being able to complete the challenges; for having reached the final challenge"; "I was pleased when I could complete the challenges".
- **Fascination:** "The storyline, the presence of a laboratory in a regular house, and the two corpses. The sequence of steps that always leads us to search for more clues"; "I was fascinated by the cockroaches and the corpse"; "I was fascinated by the game mechanics".
- **Desire:** "The desire to finish the game"; "Finding the various puzzle pieces. Wanting to open the doors of the furniture and the present on the table".
- **Sadness:** "When I found Mary's body"; "The fact that I realized the importance of language in everyday life; not being able to complete the game".
- **Fear:** "When I went to get the diary page on the balcony; entering dark areas"; "The overall dark tone and finding the corpses, and the woman's corpse following the player with her eyes"; "I felt scared when I saw the cockroaches"; "I felt anxious every time I heard the voices in my ear".
- **Anger:** "I felt frustrated initially until I understood how the game worked, like using the ax to open the trash can. Having to go back and retrieve the diary pages. I felt frustrated when I dropped the board on the balcony"; "I would get frustrated with myself when I didn't immediately complete the challenges".

The Gaming group participants also described their emotional triggers:

- **Joy:** "Solving the puzzles"; "Exploration"; "I felt joyful as soon as I understood the game controllers".
- **Admiration:** "Set dressing"; "Assuming admiration as surprise and not fascination, I was amazed when I saw the cockroaches and when I saw Mary's eyes following me"; "The technical elements of VR".
- **Pride:** "Solving the puzzles"; "Completing objectives"; "Having successfully completed the puzzles after attempts"; "Completing the puzzles, especially the one with the chest"; "I felt proud whenever I completed a challenge".
- **Hope:** "Finding clues".
- **Satisfaction:** "Solving the puzzles"; "Progression"; "Putting the keys in the locks and seeing them turn"; "Completing the challenges, and interestingly, I also felt satisfied whenever I could open a door without interruptions".
- **Fascination:** "The way the game clues were arranged"; "Discovering that I was John when I looked in the mirror"; "I was fascinated by Mary's eye movement when I approached her. I was also fascinated by the rotating mechanism of the numbers on the chest lock".
- **Desire:** "The story provoked a desire to finish the game, although I'm not sure if this is the intended interpretation of desire".
- **Sadness:** "Mary's story".
- **Fear:** "Mainly the visual and sound environment"; "Passing the board on the balcony; Mary's eyes made me uncomfortable when picking up a page"; "The eerie sounds made me quite uneasy, but the scariest moments were when I had to pass through the darkest part of the house, the stairs".
- **Embarrassment:** "Taking some time to understand what I had to do to decipher the chest code".
- **Disgust:** "Perhaps the main character, in my interpretation of the story".
- **Anger:** "The chest event, searching for the pages"; "Having to go back and forth to pick up the diary pages; the chest puzzle"; "Every time I put the keys in the wrong door and the key fell to the ground"; "In terms of anger, I'll mention frustration. I felt

slightly frustrated at the beginning when trying to understand the controllers. I also felt frustrated when I couldn't open the chest (but I admit I was completely distracted)".

- **Boredom:** "Nothing in particular, although some tasks were a bit laborious"; "Only being able to carry two things at a time in my hands".
- **Disgust:** "THE OPEN-EYED CORPSES!"; "Green liquid"; "The cockroaches in the trash made quite an impression, especially because of the sound"; "The cockroaches in the trash bin"; "CockroacheI. I was disgusted by the cockroaIs... Interestingly, the corpses didn't bother me, but the cockroacI did..".

Emotion Intensity

This section corresponds to the level of intensity experienced for each emotion. Participants provided a rating to express how intensely they felt each emotion. Figure 16 presents the average intensity value for each emotion, distinguishing between the Nursing and Gaming groups.

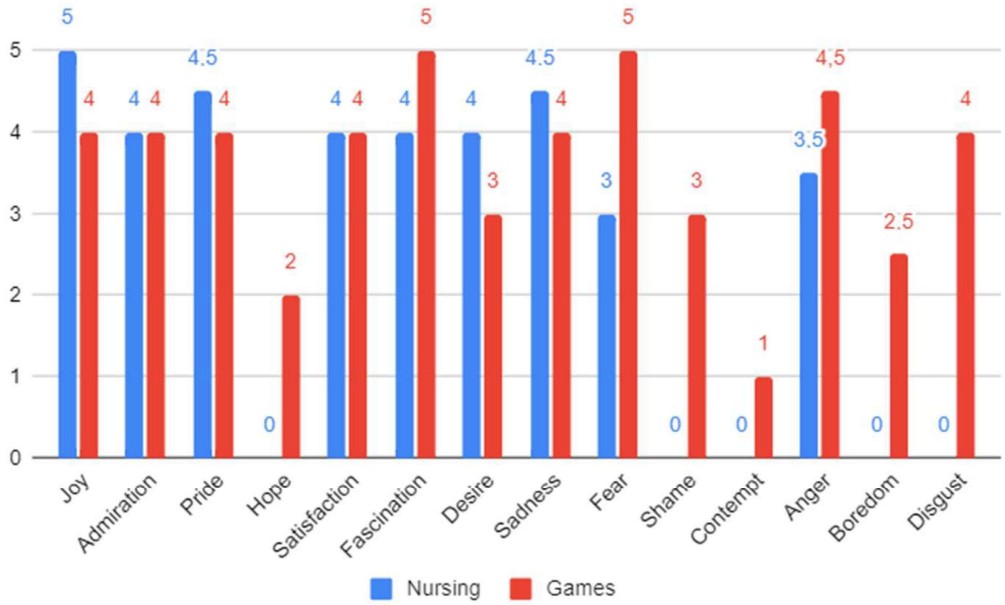

**Figure 16.** Average intensity values for emotions.

The following table (Table 6) present the emotions felt by all participants, differentiates between the Nursing and Gaming domains, and shows the result of their Confidence Interval.

**Table 6.** Standard deviation (SD) and 95% confidence interval (CI) calculated for the average intensity of each emotion felt by each group.

| Emotion | Group | Mean | SD | 95% CI |
|---|---|---|---|---|
| Joy | Nursing | 5 | 0.577 | $5 \pm 0.451$ |
| | Games | 4 | 0.894 | $4 \pm 0.699$ |
| Admiration | Nursing | 4 | 0.894 | $4 \pm 0.699$ |
| | Games | 4 | 0.894 | $4 \pm 0.699$ |
| Pride | Nursing | 4.5 | 0.577 | $4.5 \pm 0.451$ |
| | Games | 4 | 0.894 | $4 \pm 0.699$ |
| Hope | Nursing | 0 | 0.000 | $0 \pm 0.000$ |
| | Games | 2 | 0.577 | $2 \pm 0.451$ |
| Satisfaction | Nursing | 4 | 0.577 | $4 \pm 0.451$ |

**Table 6.** *Cont.*

| Emotion | Group | Mean | SD | 95% CI |
|---|---|---|---|---|
| | Games | 4 | 0.894 | 4 ± 0.699 |
| Fascination | Nursing | 4 | 0.577 | 4 ± 0.451 |
| | Games | 5 | 0.577 | 5 ± 0.451 |
| Desire | Nursing | 4 | 0.894 | 4 ± 0.699 |
| | Games | 3 | 0.577 | 3 ± 0.451 |
| Sadness | Nursing | 4.5 | 0.577 | 4.5 ± 0.451 |
| | Games | 4 | 0.894 | 4 ± 0.699 |
| Fear | Nursing | 3 | 0.577 | 3 ± 0.451 |
| | Games | 5 | 0.577 | 5 ± 0.451 |
| Shame | Nursing | 0 | 0.000 | 0 ± 0.000 |
| | Games | 3 | 0.577 | 3 ± 0.451 |
| Contempt | Nursing | 0 | 0.000 | 0 ± 0.000 |
| | Games | 1 | 0.577 | 1 ± 0.451 |
| Anger | Nursing | 3.5 | 0.577 | 3.5 ± 0.451 |
| | Games | 4.5 | 0.577 | 4.5 ± 0.451 |
| Boredom | Nursing | 0 | 0.000 | 0 ± 0.000 |
| | Games | 2.5 | 0.577 | 2.5 ± 0.451 |
| Disgust | Nursing | 0 | 0.000 | 0 ± 0.000 |
| | Games | 4 | 0.894 | 4 ± 0.699 |

User Experience

In the last section, participants answered questions related to the user experience based on the GUESS questionnaire, which is presented on a linear scale from zero as "Strongly disagree" to five as "Strongly agree". Participants provided ratings for each question, and the results are displayed in Figure 17, differentiating between the Nursing and Games groups.

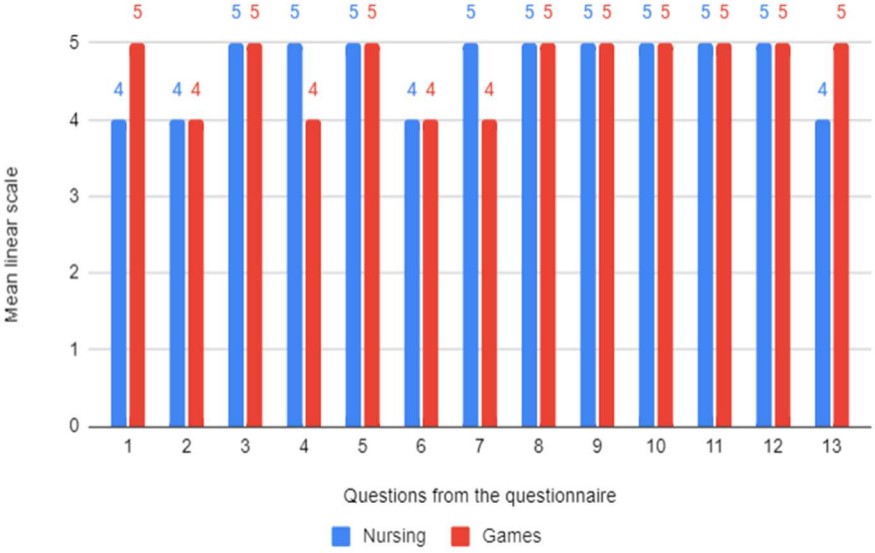

**Figure 17.** Average ratio for questionnaire's last section evaluated questions.

In the first question, "It was easy to learn how to play", four players chose option five, four players chose option four, and two players chose option three. In the second question, "I always knew what my next objective was in the game", three players chose option five, four players chose option four, one player chose option three, and two players chose option two. In the third question, "Interactions were easy to learn", six players chose option five, two players chose option four, and two players chose option three. In the fourth question, "I was engaged with the game's story from the beginning", six players chose option five, two players chose option four, one player chose option two, and one player chose option zero. In the fifth question, "I became more interested in the narrative as I progressed in the game", six players chose option five, three players chose option four, and one player chose option zero. In the sixth question, "The game's narrative was clear", two players chose option five, five players chose option four, two players chose option three, and one player chose option zero. In the seventh question, "I forgot about my daily concerns while playing", six players chose option five, two players chose option four, one player chose option three, and one player chose option zero. In the eighth question, "I had fun while playing", eight players chose option five, and two players chose option four. In the ninth question, "I felt my curiosity increasing during the game", eight players chose option five, and two players chose option four. In the tenth question, "The game is original", nine players chose option five, and one player chose option three. In the eleventh question, "The game's audio (e.g., sound effects, music) enhanced my gaming experience", seven players chose option five, one player chose option four, one player chose option three, and one player chose option zero. In the twelfth question, "The game's aesthetics (visual and sound elements) match the game's style", six players chose option five, three players chose option four, and one player chose option three. In the thirteenth question, "I think the game is visually appealing", five players chose option five, four players chose option four, and one player chose option two. Table 7, present below, presents the results of the Confidence Interval for each group based on each question provided in this section of the questionnaire.

**Table 7.** Standard deviation (SD) and 95% confidence interval (CI) calculated for each question by each group.

| Question | Group | Mean | SD | 95% CI |
| --- | --- | --- | --- | --- |
| It was easy to learn how to play | Nursing | 4 | 0.577 | $4 \pm 0.451$ |
|  | Games | 5 | 0.577 | $5 \pm 0.451$ |
| I always knew what my next objective was in the game | Nursing | 4 | 0.577 | $4 \pm 0.451$ |
|  | Games | 4 | 0.577 | $4 \pm 0.451$ |
| Interactions were easy to learn | Nursing | 5 | 0.577 | $5 \pm 0.451$ |
|  | Games | 5 | 0.577 | $5 \pm 0.451$ |
| I was engaged with the game's story from the beginning | Nursing | 4 | 0.577 | $4 \pm 0.451$ |
|  | Games | 5 | 0.577 | $5 \pm 0.451$ |
| I became more interested in the narrative as I progressed in the game | Nursing | 5 | 0.577 | $5 \pm 0.451$ |
|  | Games | 5 | 0.577 | $5 \pm 0.451$ |
| The game's narrative was clear | Nursing | 4 | 0.577 | $4 \pm 0.451$ |
|  | Games | 4 | 0.577 | $4 \pm 0.451$ |
| I forgot about my daily concerns while playing | Nursing | 5 | 0.577 | $5 \pm 0.451$ |
|  | Games | 4 | 0.577 | $4 \pm 0.451$ |

**Table 7.** *Cont.*

| Question | Group | Mean | SD | 95% CI |
|---|---|---|---|---|
| I had fun while playing | Nursing | 5 | 0.577 | 5 ± 0.451 |
| | Games | 5 | 0.577 | 5 ± 0.451 |
| I felt my curiosity increasing during the game | Nursing | 5 | 0.577 | 5 ± 0.451 |
| | Games | 5 | 0.577 | 5 ± 0.451 |
| The game is original | Nursing | 5 | 0.577 | 5 ± 0.451 |
| | Games | 5 | 0.577 | 5 ± 0.451 |
| The game's audio (e.g., sound effects, music) enhanced my gaming experience | Nursing | 5 | 0.577 | 5 ± 0.451 |
| | Games | 5 | 0.577 | 5 ± 0.451 |
| The game's aesthetics (visual and sound elements) match the game's style | Nursing | 5 | 0.577 | 5 ± 0.451 |
| | Games | 5 | 0.577 | 5 ± 0.451 |
| I think the game is visually appealing | Nursing | 4 | 0.577 | 4 ± 0.451 |
| | Games | 5 | 0.577 | 5 ± 0.451 |

Virtual Reality

Regarding the Virtual Reality part of the questionnaire, in the question "Did you experience any limitations or issues with using Virtual Reality?", one player chose option five, two players chose option four, two players chose option three, four players chose option one, and one player chose option zero. The following list presents the comments from participants explaining the limitations they felt:

- "Nausea; dizziness";
- "I felt dizzy";
- "Some initial difficulty due to never having used Virtual Reality and nausea at the end during the laboratory part, when searching for and placing pieces in the cabinet";
- "The glasses often became blurry";
- "The locomotion caused some physical discomfort, although minimal, which is rare";
- "The glasses would blur the game several times; the player's movements should be slowed down";
- "Discomfort with prolonged use";
- "As I wear glasses, I had to tighten the Quest to my face, which started to hurt my forehead after some time";
- "I only had problems at the beginning to calibrate the glasses, but it was a minor issue. I had some limitation in opening the lock of the chest, but it was my fault because I was distracted".
- Feedback

To conclude the questionnaire, an optional question, "Do you have any doubts or suggestions regarding the game?" was asked, and three participants responded. Here is a list of the answers given by the players:

- "Having a Portuguese version";
- "Slowing down the player's movement; the game should be shorter; there was no direct representation of some emotions, and others were very similar";
- "While the darkness adds to the atmosphere, the initial area has unnecessarily dark areas, in my o".

4.2.2. Focus Group Results

After participants completed the questionnaire, they were asked to participate in a focus group where we could have an open conversation about their experience during the game. All participants agreed.

During the focus group, most participants mentioned that at the beginning they were having difficulty understanding how to play the game, mainly because they had never experienced Virtual Reality or played an Escape Room before. Some participants mentioned that they spent a lot of time looking at objects, waiting for something to happen. Despite this, all participants mentioned that once they understood how to play, it was much easier to continue the game and enjoy it. Until they grasped the game's logic, no player progressed beyond the kitchen, so they mentioned that it did not interfere with the gaming experience.

After getting into the game's flow, almost all participants mentioned that they forgot about the real world and were completely immersed in the virtual environment. Some barriers to this immersion were: (1) not understanding English and needing external help, and (2) external noise in the university hallways. Additionally, some participants mentioned that they did not adjust the Oculus Quest properly to their faces, causing the headset to move and blur the game during certain movements, reminding them that they were in a virtual environment.

Regarding player movement in the virtual world, some participants mentioned that it was challenging to adjust the movement speed according to what they wanted to do. A practical example was when they wanted to approach an object, and instead of applying gentle pressure on the joystick or physically moving in the real world, they pushed the joystick to the maximum. This caused motion sickness as they moved too quickly towards an object and were suddenly pushed backward. The same participants mentioned feeling reluctant to move in the real world for fear of hitting a wall, despite the virtual barrier when approaching physical objects.

Regarding the elicitation of emotions, all participants mentioned that the game achieved its objective. They reported feeling some form of emotion throughout the game, even if it was just anticipatory anxiety. Regarding the emotion of anger, some participants mentioned not feeling anger but rather a sense of frustration. Furthermore, regarding anger and frustration with the puzzles, six participants mentioned that they expected the puzzles to be more difficult, so they were looking for a more challenging and less obvious solution. This phenomenon led to frustration not only with the puzzle but also with themselves. As for the emotion of fear, one participant specifically mentioned that they had never considered themselves afraid of heights, but when they had to cross the plank on the balcony, it caused intense fear, sweaty hands, and trembling throughout their body.

With this comment, several participants mentioned that they also had physical reactions throughout the experience, including trembling, sweating, and an accelerated heart rate. These physical reactions were reported by participants as keeping them more alert during the game, as a defense mechanism triggered by fear. Regarding fear, another participant mentioned that in real life, they are not bothered by darkness, but in the virtual environment, they felt quite anxious when passing through dark areas. In addition to these aspects, the flashlight was mentioned as an element that caused fear. Although no participant mentioned this element as a trigger in the questionnaire, during the focus group, participants mentioned being afraid of losing the flashlight. This fear stemmed from the concern of not being able to find it again and the uncertainty of what might happen in the darkness.

Regarding the game's narrative, half of the participants did not have the opportunity to complete the game and reach the end of the narrative. Despite this, two predominant theories emerged about what happened. The first theory was that John had committed suicide due to what happened to Mary. The second theory was that John had killed Mary and was fabricating an entire story. Some players mentioned that they liked the open-ended nature of the story, as it allowed them to fill in the gaps and immerse themselves more deeply in the experience. However, unlike other players, they felt that many details were

missing from the narrative, and they did not develop the creativity to interpret the narrative from a personal perspective.

Regarding motion sickness, apart from how players chose to move, a factor that contributed to players experiencing discomfort was the delay in their movement initiation. Sometimes it seemed like they were in slow motion, taking time to move forward, and then suddenly moving quickly. This occurred because we were sharing the game screen on the computer to assist participants when needed. However, this issue only affected the first six participants, so we decided to stop screen sharing for a better experience. Since the following players had experience in VR and understood English, there was no need for screen sharing.

Overall, all participants enjoyed the game and had a positive experience. They felt that more negative emotions were elicited than positive ones, but this did not hinder the experience; on the contrary, it added to the overall in.

### 4.2.3. Evaluation of Results and Feedback

After the questionnaire and focus group results, it was possible to assess the game in terms of eliciting emotions, visual and sound aspects, narrative, and user experience.

Starting with the elicitation of emotions, it was clear that the goal of the game was achieved by provoking numerous emotions in the ten participants. Comparing the list of game elements designed to evoke positive and negative emotions with the list of game elements mentioned by the players and their corresponding triggers, it can be seen that the intended triggers did indeed provoke emotions. Additionally, there were some additional elements that were mentioned by the players. Table 8 presents the triggers designed by the team, along with the additional elements mentioned by the players and the corresponding emotions felt.

**Table 8.** Triggers/emotions description.

| Triggers | Emotions |
| --- | --- |
| Completing objectives | Joy, Pride, Satisfaction |
| Progressing in the game | Pride, Satisfaction |
| Game mechanics | Admiration, Satisfaction, Fascination |
| Exploration | Joy |
| Visual aspect | Admiration |
| Finding clues | Hope, Fascination |
| Finding pieces | Desire |
| Narrative | Fascination, Desire, Sadness, Contempt |
| Easy-to-solve puzzles | Anger |
| Lack of inventory | Boredom |
| Dark environment | Fear |
| Flashlight | Fear |
| Cockroaches | Admiration, Fascination, Disgust |
| Chest lock puzzle | Pride, Fascination, Shame, Anger |
| Mary's body and eyes | Admiration, Fascination, Sadness, Fear, Disgust |
| Sound design | Fear |
| Height | Fear |
| Keys in locks | Satisfaction, Anger |
| John's body | Admiration, Fear, Disgust |
| Toxic liquid | Disgust |

As one can see from Table 8, in addition to the planned triggers, there were other triggers reported by the players. Through Table 8, we can also observe that several elements planned to evoke negative emotions also elicited positive emotions simultaneously. Although several triggers caused anger in players, the fun elements of completing objectives or solving puzzles were often accompanied by satisfaction [5,15]. The fact that the puzzles were easy to solve led to feelings of anger and frustration when players were unable to solve them. This phenomenon is consistent with research on Escape Rooms, where it was found that while developing a creative mindset for puzzle-solving is beneficial, it can also lead to limitations as players expect the solutions to be more complex than they actually are [12]. Furthermore, players' frustration with themselves stems from three fundamental concepts of VR: interaction, immersion, and engagement [10,11]. This is because an active elicitation of emotions was employed, where participants actively interacted with the environment that was designed to provoke emotions [10].

None of the participants mentioned the need to memorize symbols and letters to solve two puzzles. As we observed the players during the game, we noticed that they spent some time memorizing the symbols and letters, and some players even recited them as they progressed. Five participants had to go back to review the symbols and letters. This phenomenon can be attributed to stress, which affects players' memory and poses limitations for puzzles with solutions spread across different rooms [12]. In summary, the game successfully achieved its objective of eliciting a variety of emotions through a wide range of triggers.

The visual aspect of the game was highly commented on by the participants, both positively and negatively. Players greatly appreciated the visual design of the house and its decorations. However, they also expressed fear in the dark areas, suggesting that they might have been excessively dark. We cannot fully assess the validity of this comment, as it is uncertain whether it hinders gameplay or simply serves its purpose of making players uncomfortable in those areas.

Regarding the sound design of the game, only the most important sounds were included. Sounds such as objects falling were not added in this prototype phase, although none of the players mentioned their absence. Based on the questionnaire responses, it is evident that the most prominent sounds were the whispers in the dark area on the stairs, which evoked fear, anxiety, and apprehension in the players. Only one participant rated the sound design a zero, attributing it to the significant background noise in the university that prevented them from hearing any in-game sounds.

Regarding the game's narrative, only five players had the opportunity to complete the game and experience the full narrative. However, all participants were briefed on the narrative during the focus group session. Only one participant rated all aspects related to the narrative with zero, simply because they were unable to progress far enough in the game to form an opinion. After analyzing the questionnaire responses and considering the comments from the focus group, it can be concluded that the successfully fulfilled its purpose and conveyed the intended messages. Although some players felt that the narrative was incomplete, this was the intended effect, allowing players to use their creativity and imagination to fill in the gaps. Based on research on Escape Rooms, involving players in the narrative is crucial for enhancing their experience, and one effective approach is to leave the story incomplete to stimulate players' imagination and further engage them in the experience [14]. Therefore, the open-ended narrative was generally a good choice and achieved its objective for most Ints.

The user's experience was evaluated through thirteen questions based on the GUESS questionnaire. The results indicate that the game, as an initial prototype, met expectations. However, there are several aspects to consider for future development: (1) adjusting the physical space to be smaller, (2) improving movement mechanics, (3) incorporating a tutorial scene or on-screen instructions for interactions, (4) providing a Portuguese version, (5) creating a clearer objective scheme, and (6) enhancing visual feedback.

The adjustment of physical space is necessary because the current space is too long, and prolonged gameplay caused some participants to feel nauseous. Additionally, for classroom use with our partners, the game should be playable by any type of player within a maximum of thirty minutes from start to finish. Improving movement mechanics will also help reduce motion sickness and accelerate gameplay. Although joystick movement provides greater freedom, as discussed in the work by Samira and Layla, teleportation is a more effective movement method to mitigate motion sickness by minimizing the disparity between the virtual and real worlds [7]. Some players experienced this disparity, especially when screen sharing the game with a computer. While teleportation movement reduces gameplay time, it also limits the exploration opportunities that can be provided to players. Consequently, the physical space needs to be larger if players decide to explore or interact closely with objects.

The idea of creating a tutorial scene or providing on-screen instructions for button prompts is necessary because many future players will be unfamiliar with Virtual Reality. This approach allows each player to learn how interactions work and familiarize themselves with the controls without interfering with gameplay or the overall experience.

The inclusion of a Portuguese version of the game was already planned among all partners to ensure that future players participating in classes at the University of Minho can complete the game without breaking immersion by seeking external assistance. This element was scheduled for implementation after the development of this prototype.

Creating a clearer objective scheme stems from the fact that simple on-screen text may not effectively capture players' attention. All participants expressed the need to frequently press the 'A' button to review information.

The need for improved visual feedback was evident from players' comments regarding the puzzle lock and key interactions. Although players experienced positive emotions upon completing these objectives and challenges, from external observation while they played, they could not immediately understand why nothing was happening.

Regarding the difference between players from the nursing and game design fields, it was understood that players with more gaming experience found it easier to complete the game. Although there were participants from the Nursing field with some gaming experience, the more detailed and professional knowledge related to gaming experiences enabled the game design participants to perform better and complete puzzles more efficiently. Additionally, as the game design participants have more experience with various types of games, they experienced more anger with certain game elements compared to the nursing participants. This is because in entertainment games, such emotions are not intentionally provoked in players, mainly because it can disengage them. In this group of nursing students, only one person had previous experience with Virtual Reality, but they mentioned being accompanied by their child and never having played something as complex as this game. The lack of Virtual Reality experience was evident in both the gameplay time and the occurrence of motion sickness. Despite these differences, both fields demonstrated their ability to play the game and enjoyed it, making the game suitable for players of any background.

The 95% confidence intervals for the "Emotions Experienced" section indicate that the reported emotions were quite consistent within both the Nursing and Gaming groups. This suggests that the Virtual Reality Escape Room experience effectively elicited a range of emotions in participants from both backgrounds, demonstrating the game's capacity to provoke emotional responses irrespective of prior professional exposure. In the "Emotion Intensity" section, the results are notably positive. Both groups displayed similar mean intensity scores for various emotions, indicating that the game maintained a consistent emotional impact regardless of the participant's background. These findings affirm the game's effectiveness in consistently evoking emotions, further supporting its capacity for engaging players from diverse fields. The "User Experience" results, characterized by the 95% confidence intervals, reveal consistently high scores across all evaluated aspects for both groups. This implies that the game was well-received by participants from both

the Nursing and Gaming backgrounds. The high ratings signify that the game effectively engaged players and created an immersive and enjoyable experience, showcasing its potential to cater to a broad audience. Overall, these confidence intervals reflect positively on the game's effectiveness in delivering a satisfying user experience across different participant profiles.

## 5. Conclusions

### 5.1. Main Discoveries and Contributions

For the development of this game, the first investigation focused on emotions and the human being. In this research, we learned that there is no defined concept of emotion, and there is a high probability that researchers in this field will never reach a consensus. Emotions are highly complex and can be approached from various perspectives. Throughout the historical journey, we learned that several researchers and philosophers created their own theories about emotions and how they manifest in human beings. During the research for this project, we relied on the work of two main researchers: Paul Ekman [25] and Norman [26]. From Paul Ekman, we learned that emotions can be interpreted through facial expressions. Ekman developed a measurement system for identifying emotions based on generic facial expressions from different individuals. Thanks to this, we decided to use PrEmo images in the questionnaire to represent emotions, accompanied by text for easier data analysis. With Norman, we learned how the human body processes emotions and how it reacts to emotional stimuli. Additionally, Norman explains in a practical way how a negative emotional stimulus can also evoke a positive emotion. Studying all the information in this section made it clear that there is no way to predict which exact emotion the players will feel. Therefore, we relied on Norman's insights to create a table of positive and negative emotions, along with their corresponding game elements as triggers. This research confirmed hypothesis one, which predicted that it is impossible to predict the exact emotion a player will feel.

The major advantage of using VR is that there is no barrier between the game character and the player, creating a sense of being fully immersed in the game world. This active elicitation of emotions allows players to actively participate in generating their emotions. It leads to a high level of immersion, which makes players feel like they are truly in the virtual world, disconnecting from the real world and stimulating their senses for a more engaging experience. VR was considered by us a valuable tool and a direct means for eliciting emotions due to its proximity to reality. Based on the test results, it was evident that players felt like everything was happening to them in reality. All participants became absorbed in the game and completely focused on it. This research and the results confirmed hypothesis three, which predicted that VR contributes to a better experience.

We chose the Escape Room genre after conducting some research because it is a game type designed to evoke emotions in players, usually negative emotions such as fear and anger, followed by positive emotions such as pride and satisfaction. Based on the results of our tests, we observed that Escape Room was undoubtedly the best choice for this theme, and everything we anticipated happened. Despite the intensive provocation of negative emotions, players continued to enjoy the game and never considered giving up. These results confirmed hypotheses four and five. From this, we can conclude that the elicitation of negative emotions does not make players quit the game or dislike it.

The phase of testing with the two different groups occurred after the completion of the entire prototype development. These two groups of participants were divided into five from the nursing field and five from the game development field. This testing phase was a success, and we evaluated the game as effective in eliciting emotions from players, fulfilling its objective. Based on the test results and the evaluation presented in Section 4 of this paper, we were able to validate hypotheses two, six and seven. Regarding hypothesis six, we can learn that participants' feelings of anger with the chest lock puzzle led them to spend more time trying the same code multiple times. Another example of this hypothesis was seen when participants had to cross the plank on the balcony to retrieve a diary page.

Many participants approached this challenge with caution and attention due to fear or concern about falling. Hypothesis seven, which predicted that the same game element could evoke multiple positive and negative emotions in different individuals, was observed in many cases. One example of this phenomenon was with Mary's eyes, as players were both startled by the eyes following their movements and fascinated by the effect. Regarding hypothesis two, despite participants experiencing the anger emotion, some did not select it in the questionnaire because it was followed by pride or satisfaction. A practical example was the puzzle for the chest lock, where almost all participants understood the correct year but spent a significant amount of time trying the same year repeatedly until they realized that a date is more than just the year. In this sense, despite feeling frustration, the positive emotion of accomplishment prevailed over the negative one when they successfully completed the puzzle. In addition to these aspects, the flashlight was also mentioned as an element that caused fear in the focus group, but no participant identified this element as a trigger in the questionnaire.

In conclusion, we can verify that the prototype was a success among our participants and partners. Although the game "Escape VR: The Guilt" already appears quite solid, there are areas that can be improved. All these potential enhancements are presented in the next section.

*5.2. Comparison to Previous Studies*

In the realm of VR gaming and emotion elicitation, our recent study offers a notable expansion of our understanding compared to previous works such as the XEODesign study and the VR Escape Room Game analysis. One of the standout features of our research lies in its comprehensive exploration of a wide range of emotions experienced by players within VR Escape Room settings. While XEODesign focused on emotions in a variety of games our study delved into a diverse emotional landscape in a specific genre. Like the VR Escape Room Game, we used negative emotions along with the use of puzzles and environment. Despite the similarity with this study, they do not conduct a deep emotional analysis in their study. Our meticulous examination of various emotions provides a nuanced understanding of players' emotional responses, allowing for a more intricate analysis of VR gaming experiences.

Furthermore, our study assessed user experience in VR Escape Room games with a heightened level of depth. While both the XEODesign and VR Escape Room Game analyses evaluated user experience, our research stood out by implementing questions based on the GUESS questionnaire. Our questionnaire offers a structured and comprehensive approach to gauging various aspects of player engagement, immersion, and overall experience. By employing this method, our study provided richer insights into how players perceive and engage with VR Escape Room games, contributing to a more holistic evaluation of user experiences.

Another distinctive feature of our project was the diversity within our participant groups. While the XEODesign study and the VR Escape Room Game analysis focused on specific player profiles, our study incorporated participants from two backgrounds, including individuals from the Nursing and Gaming domains, who may or may not be familiar with games and/or VR. Despite the small number of participants, this sample allowed for a broader assessment of how different individuals, with varying experiences and expectations, respond emotionally and experientially to VR Escape Room games.

Moreover, our study introduced a rigorous statistical dimension that was not present in previous studies. Through the utilization of standard deviation and confidence intervals, our study enhanced the credibility and robustness of its findings. This incorporation of quantitative methods ensured that the emotional responses and user experiences within VR Escape Room games were evaluated with a high degree of scientific rigor.

Lastly, our study ventured beyond merely identifying emotions and delved into the specific game elements responsible for triggering these emotional responses. By conducting this granular analysis, our research provides actionable insights for VR game designers. It

offers guidance on how to effectively craft VR Escape Room experiences that elicit the desired emotional reactions, thereby enhancing the overall gameplay and player engagement.

In conclusion, our study significantly advances the understanding of emotional engagement within VR Escape Room games. By exploring a broader spectrum of emotions, utilizing comprehensive questionnaires, incorporating diverse participant profiles, integrating statistical rigor, and analyzing the precise triggers of emotions, our research makes a substantial contribution to the existing body of literature. It not only enriches the comprehension of emotional experiences in VR gaming but also offers valuable insights for the design and development in future VR Escape Room games.

*5.3. Future Improvements and Recommendations*

The creation of "Escape VR: The Guilt" demonstrated both substantial successes and places for future growth in light of the evaluation results. The game was successful in evoking a variety of feelings in the players, demonstrating good emotion elicitation. While intended triggers were successful in achieving their objectives, players also had additional emotional reactions, which showed a dynamic gameplay experience. To improve puzzle-solving, certain factors, such as the requirement for memorization and limits brought on by stress, need more attention.

There was disagreement on the sound and visual designs. Although many players complimented the fascinating graphics and interesting atmosphere, several had worries about the game's extremely gloomy spots. Participants reported feeling scared and anxious thanks to the sound design; however, one player's experience was hampered by background noise. To achieve the ideal combination of comfort and immersion, these factors should be adjusted. Despite being unfinished, the game's narrative successfully delivered its intended themes and captured players' imaginations, increasing their immersion in the experience. However, addressing the unfinished story and providing a more thoughtful conclusion can increase player interest. The initial prototype's user experience was satisfactory, but it also highlighted certain areas that needed improvement. Motion sickness can be reduced, and gameplay can be made more efficient by condensing the physical environment and adding more logical movement mechanics. It is essential to provide a tutorial or on-screen instructions to help gamers, especially those new to VR, become accustomed to the controls and interactions. The Portuguese translation will meet the needs of the University of Minho for classroom use. To successfully assist players and increase overall gameplay enjoyment, a clearer objective structure and better visual feedback are necessary. These changes will guarantee a more seamless and engaging gameplay experience. Additionally, it is important to recognize the contrasts between players in the domains of nursing and game design. The intensity and challenges of the game can be adjusted to best suit the diverse skill and experience levels of the players.

In conclusion, "Escape VR: The Guilt" has successfully elicited emotions and engaged participants, laying a good foundation for an entertaining VR gaming experience. The game will develop and maybe undergo further versions if the suggested improvements are addressed and the feedback from players with various backgrounds is taken into consideration. Continued research and development can enhance the VR gaming scene as the gaming industry develops, delivering an engaging and pleasurable experience for gamers of all ages and ability levels.

**Author Contributions:** Conceptualization, I.O. and L.G.; methodology, I.O., V.C., F.S., P.N., E.O. and L.G..; software, I.O.; validation, V.C., F.S., P.N., E.O. and L.G.; investigation, I.O.; resources, I.O., E.O. and L.G.; writing—original draft preparation, I.O. and V.C.; writing—review and editing, I.O., V.C., F.S., P.N., E.O. and L.G.; visualization, I.O., V.C., F.S., P.N., E.O. and L.G.; supervision, E.O. and L.G.; project administration, L.G.; funding acquisition, L.G. and V.C. All authors have read and agreed to the published version of the manuscript.

**Funding:** This paper was funded by Erasmus+ program (Project Nº 2020-1-PT01-KA203-078847) and through the FCT/MCTES under the projects UIDB/05549/2020 and UIDP/05549/2020.

**Data Availability Statement:** Not Applicable.

**Conflicts of Interest:** The authors declare no conflict of interest.

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
