# Peer review of "Development of a Virtual Reality Escape Room Game for Emotion Elicitation"

_information, doi:10.3390/info14090514_

Round 1

Reviewer 1 Report

This study aims at developing and testing a Virtual Reality Escape Room Game for Emotions Elicitation.

The idea of this study is interesting with significant potential to contribute to the discussion about the potential of VR gaming technologies to elicit emotions. The gaming characteristics indicate that the researchers have done responsible research.

Some revisions concern mainly the way this study is presented and organized.

The abstract needs a minor revision, strengthening its structure and highlighting the purpose of the study, the methodology used, the main findings, and the conclusions or interpretations.

The introduction places the study in a broad context and adequately describes the objectives. It would be useful to conclude the introduction briefly highlighting the potential of this study.

I believe that the section “context and importance”, although it includes several strong arguments (ie the importance of emotions in increasing engagement during gaming), seems weaker as a whole possibly because of the way arguments are structured and the references used.

Throughout the text, several statements need relevant references. For instance in the conclusions.

Conclusions, although extensive, are interesting with several critical interpretations.

The title “Literature review” in the second section is necessary?

Figures are useful. If possible, to improve the quality of some pictures (ie figure 3).

The reference list should follow the ACS style guide.

Language is understandable. However, minor revisions are required to improve the flow and facilitate the reader to get into the text and follow step by step the research process. 

Reviewer 2 Report

The article describes the effectiveness of eliciting emotions in Escape Room Game by utilizing Virtual Reality (VR) technology.

Though there are lack of novelty in academic aspect in this article, the article provides valuable empirical design points for VR game developers.

To improve the article, I suggest the following:

1. Measurement of the effectiveness is only measured by questionnaire results which can be subjective and/or biased depending on their experimental setups.

The authors must give confidence level or similar in statistical aspect.

2.

There are many similar research works on emotion elicitation in VR games.

So, it is recommended to compare the experimental results or 'Escape VR: The Guilt' game with several other similar works and describe similar and distinctive points, and relative pros and cons of the suggestion.
